# SGD LEARNS ONE-LAYER NETWORKS IN WGANS

## ABSTRACT

Generative adversarial networks (GANs) are a widely used framework for learning generative models. Wasserstein GANs (WGANs), one of the most successful variants of GANs, require solving a minmax optimization problem to global optimality, but are in practice successfully trained using stochastic gradient descent-ascent. In this paper, we show that, when the generator is a one-layer network, stochastic gradient descent-ascent converges to a global solution with polynomial time and sample complexity.

## 1 INTRODUCTION

Generative Adversarial Networks (GANs) (Goodfellow et al., 2014) are a prominent framework for learning generative models of complex, real-world distributions given samples from these distributions. GANs and their variants have been successfully applied to numerous datasets and tasks, including image-to-image translation (Isola et al., 2017), image super-resolution (Ledig et al., 2017), domain adaptation (Tzeng et al., 2017), probabilistic inference (Dumoulin et al., 2016), compressed sensing (Bora et al., 2017) and many more. These advances owe in part to the success of Wasserstein GANs (WGANs) (Arjovsky et al., 2017; Gulrajani et al., 2017), leveraging the neural net induced integral probability metric to better measure the difference between a target and a generated distribution.

Along with the afore-described empirical successes, there have been theoretical studies of the statistical properties of GANs—see e.g. (Zhang et al., 2018; Arora et al., 2017; 2018; Bai et al., 2018; Dumoulin et al., 2016) and their references. These works have shown that, with an appropriate design of the generator and discriminator, the global optimum of the WGAN objective identifies the target distribution with low sample complexity.

On the algorithmic front, prior work has focused on the stability and convergence properties of gradient descent-ascent (GDA) and its variants in GAN training and more general min-max optimization problems; see e.g. (Nagarajan & Kolter, 2017; Heusel et al., 2017; Mescheder et al., 2017; 2018; Daskalakis et al., 2017; Daskalakis & Panageas, 2018a;b; Gidel et al., 2019; Liang & Stokes, 2019; Mokhtari et al., 2019; Jin et al., 2019; Lin et al., 2019) and their references. It is known that, even in min-max optimization problems with convex-concave objectives, GDA may fail to compute the min-max solution and may even exhibit divergent behavior. Hence, these works have studied conditions under which GDA converges to a globally optimal solution under a convex-concave objective, or different types of locally optimal solutions under nonconvex-concave or nonconvex-nonconcave objectives. They have also identified variants of GDA with better stability properties in both theory and practice, most notably those using negative momentum.

In the context of GAN training, Feizi et al. (2017) show that for WGANs with a linear generator and quadratic discriminator GDA succeeds in learning a Gaussian using polynomially many samples in the dimension. In the same vein, we are the first to our knowledge to study the global convergence properties of stochastic GDA in the GAN setting, and establishing such guarantees for non-linear generators. In particular, we study the WGAN formulation for learning a single-layer generative model with some reasonable choices of activations including tanh, sigmoid and leaky ReLU.

**Our contributions.** For WGAN with a one-layer generator network using an activation from a large family of functions and a quadratic discriminator, we show that stochastic gradient descent-ascent learns a target distribution using polynomial time and samples, under the assumption that the target distribution is realizable in the architecture of the generator. This is achieved by a) analysis of the dynamics of stochastic gradient-descent to show it attains a global optimum of the minmax problem, and b) appropriate design of the discriminator to ensure a parametric $\mathcal{O}(\frac{1}{\sqrt{n}})$ statistical rate (Zhang et al., 2018; Bai et al., 2018).

**Related Work.** We briefly review relevant results in GAN training and learning generative models:

- *Optimization viewpoint.* For standard GANs and WGANs with appropriate regularization, Nagarajan & Kolter (2017), Mescheder et al. (2017) and Heusel et al. (2017) establish sufficient conditions to achieve local convergence and stability properties for GAN training. At the equilibrium point, if the Jacobian of the associated gradient vector field has only eigenvalues with negative real-part at the equilibrium point, GAN training is verified to converge locally for small enough learning rates. A follow-up paper by (Mescheder et al., 2018) shows the necessity of these conditions by identifying a prototypical counterexample that is not always locally convergent with gradient descent based GAN optimization. However, the lack of global convergence prevents the analysis to provide any guarantees of learning the real distribution.

The work of (Feizi et al., 2017) described above has similar goals as our paper, namely understanding the convergence properties of basic dynamics in simple WGAN formulations. However, they only consider linear generators, which restrict the WGAN model to learning a Gaussian. Our work goes a step further, considering WGANs whose generators are one-layer neural networks with a broad selection of activations. We show that with a proper gradient-based algorithm, we can still recover the ground truth parameters of the underlying distribution.

More broadly, WGANs typically result in nonconvex-nonconcave min-max optimization problems. In these problems, a global min-max solution may not exist, and there are various notions of local min-max solutions, namely local min-local max solutions Daskalakis & Panageas (2018b), and local min solutions of the max objective Jin et al. (2019), the latter being guaranteed to exist under mild conditions. In fact, Lin et al. (2019) show that GDA is able to find stationary points of the max objective in nonconvex-concave objectives. Given that GDA may not even converge for convex-concave objectives, another line of work has studied variants of GDA that exhibit global convergence to the min-max solution Daskalakis et al. (2017); Daskalakis & Panageas (2018a); Gidel et al. (2019); Liang & Stokes (2019); Mokhtari et al. (2019), which is established for GDA variants that add negative momentum to the dynamics. While the convergence of GDA with negative momentum is shown in convex-concave settings, there is experimental evidence supporting that it improves GAN training (Daskalakis et al., 2017; Gidel et al., 2019).

- *Statistical viewpoint.* Several works have studied the issue of mode collapse. One might doubt the ability of GANs to actually learn the distribution vs just memorize the training data (Arora et al., 2017; 2018; Dumoulin et al., 2016). Some corresponding cures have been proposed. For instance, Zhang et al. (2018); Bai et al. (2018) show for specific generators combined with appropriate parametric discriminator design, WGANs can attain parametric statistical rates, avoiding the exponential in dimension sample complexity (Liang, 2018; Bai et al., 2018; Feizi et al., 2017).

Recent work of Wu et al. (2019) provides an algorithm to learn the distribution of a single-layer ReLU generator network. While our conclusion appears similar, our focus is very different. Our paper targets understanding when a WGAN formulation trained with stochastic GDA can learn in polynomial time and sample complexity. Their work instead relies on a specifically tailored algorithm for learning truncated normal distributions Daskalakis et al. (2018).

## 2 PRELIMINARIES

We consider GAN formulations for learning a generator $G_A : \mathbb{R}^k \to \mathbb{R}^d$ of the form $\boldsymbol{z} \mapsto \boldsymbol{x} = \phi(A\boldsymbol{z})$, where $A$ is a $d \times k$ parameter matrix and $\phi$ some activation function. We consider discriminators $D_{\boldsymbol{v}} : \mathbb{R}^d \to \mathbb{R}$ or $D_V : \mathbb{R}^d \to \mathbb{R}$ that are linear or quadratic forms respectively for the different purposes of learning the marginals or the joint distribution. We assume latent variables $\boldsymbol{z}$ are sampled from the normal $\mathcal{N}(0, I_{k \times k})$, where $I_{k \times k}$ denotes the identity matrix of size $k$. The real/target distribution outputs samples $\boldsymbol{x} \sim \mathcal{D} = G_{A^*}(\mathcal{N}(0, I_{k_0 \times k_0}))$, for some ground truth parameters $A^*$, where $A^*$ is $d \times k_0$, and we take $k \geq k_0$ for enough expressivity, taking $k = d$ when $k_0$ is unknown.

The Wasserstain GAN under our choice of generator and discriminator is naturally formulated as:

$$\min_{A \in \mathbb{R}^{d \times k}} \max_{\boldsymbol{v} \in \mathbb{R}^d} \left\{ f(A, \boldsymbol{v}) \equiv \mathbb{E}_{\boldsymbol{x} \sim \mathcal{D}} D_{\boldsymbol{v}}(\boldsymbol{x}) - \mathbb{E}_{\boldsymbol{z} \sim \mathcal{N}(0, I_{k \times k})} D_{\boldsymbol{v}}(G_A(\boldsymbol{z})) \right\}.^1$$

---

[1] We will replace $\boldsymbol{v}$ by $V \in \mathbb{R}^{d \times d}$ when necessary.

We use $\boldsymbol{a}_i$ to denote the $i$-th row vector of $A$. We sometimes omit the 2 subscript, using $\|\boldsymbol{x}\|$ to denote the 2-norm of vector $\boldsymbol{x}$, and $\|X\|$ to denote the spectral norm of matrix $X$. $\mathbb{S}^n \subset \mathbb{R}^{n \times n}$ represents all the symmetric matrices of dimension $n \times n$. We use $Df(X_0)[B]$ to denote the directional derivative of function $f$ at point $X_0$ with direction $B$: $Df(X_0)[B] = \lim_{t \to 0} \frac{f(X_0 + tB) - f(X_0)}{t}$.

## 3 WARM-UP: LEARNING THE MARGINAL DISTRIBUTIONS

As a warm-up, we ask whether a simple linear discriminator is sufficient for the purposes of learning the marginal distributions of all coordinates of $\mathcal{D}$. Notice that in our setting, the $i$-th output of the generator is $\phi(x)$ where $x \sim \mathcal{N}(0, \|\boldsymbol{a}_i\|^2)$, and is thus solely determined by $\|\boldsymbol{a}_i\|_2$. With a linear discriminator $D_{\boldsymbol{v}}(\boldsymbol{x}) = \boldsymbol{v}^\top \boldsymbol{x}$, our minimax game becomes:

$$\min_{A \in \mathbb{R}^{d \times k}} \max_{\boldsymbol{v} \in \mathbb{R}^d} \left\{ f_1(A, \boldsymbol{v}) \equiv \mathbb{E}_{\boldsymbol{x} \sim \mathcal{D}} \left[ \boldsymbol{v}^\top \boldsymbol{x} \right] - \mathbb{E}_{\boldsymbol{z} \sim \mathcal{N}(0, I_{k \times k})} \left[ \boldsymbol{v}^\top \phi(A\boldsymbol{z}) \right] \right\}. \tag{1}$$

Notice that when the activation $\phi$ is an odd function, such as the tanh activation, the symmetric property of the Gaussian distribution ensures that $\mathbb{E}_{\boldsymbol{x} \sim \mathcal{D}}[\boldsymbol{v}^\top \boldsymbol{x}] = 0$, hence the linear discriminator in $f_1$ reveals no information about $A^*$. Therefore specifically for odd activations (or odd plus a constant activations), we instead use an adjusted rectified linear discriminator $D_{\boldsymbol{v}}(\boldsymbol{x}) \equiv \boldsymbol{v}^\top R(\boldsymbol{x} - C)$ to enforce some bias, where $C = \frac{1}{2}(\phi(x) + \phi(-x))$ for all $x$, and $R$ denotes the ReLU activation. Formally, we slightly modify our loss function as:

$$\bar{f}_1(A, \boldsymbol{v}) \equiv \mathbb{E}_{\boldsymbol{x} \sim \mathcal{D}} \left[ \boldsymbol{v}^\top R(\boldsymbol{x} - C) \right] - \mathbb{E}_{\boldsymbol{z} \sim \mathcal{N}(0, I_{k \times k})} \left[ \boldsymbol{v}^\top R(\phi(A\boldsymbol{z}) - C) \right]. \tag{2}$$

We will show that we can learn each marginal of $\mathcal{D}$ if the activation function $\phi$ satisfies the following.

**Assumption 1.** *The activation function $\phi$ satisfies either one of the following:*
*1. $\phi$ is an odd function plus constant, and $\phi$ is monotone increasing;*
*2. The even component of $\phi$, i.e. $\frac{1}{2}(\phi(x) + \phi(-x))$, is positive and monotone increasing on $x \in [0, \infty)$.*

**Remark 1.** *All common activation functions like (Leaky) ReLU, tanh or sigmoid function satisfy Assumption 1.*

**Lemma 1.** *Suppose $A^* \neq 0$. Consider $f_1$ with activation that satisfies Assumption 1.2 and $\bar{f}_1$ with activation that satisfies Assumption 1.1. The stationary points of such $f_1$ and $\bar{f}_1$ yield parameters $A$ satisfying $\|\boldsymbol{a}_i\| = \|\boldsymbol{a}_i^*\|, \forall i \in [d]$.*

To bound the capacity of the discriminator, similar to the Lipschitz constraint in WGAN, we regularize the discriminator. For the regularized formulation we have:

**Theorem 1.** *In the same setting as Lemma 1, alternating gradient descent-ascent with proper learning rates on*

$$\min_A \max_{\boldsymbol{v}} \{ f_1(A, \boldsymbol{v}) - \|\boldsymbol{v}\|^2/2 \} \quad \text{or respectively} \quad \min_A \max_{\boldsymbol{v}} \{ \bar{f}_1(A, \boldsymbol{v}) - \|\boldsymbol{v}\|^2/2 \}$$

*recovers $A$ such that $\|\boldsymbol{a}_i\| = \|\boldsymbol{a}_i^*\|, \forall i \in [d]$.*

All the proofs of the paper can be found in the appendix. We show that all local min-max points in the sense of (Jin et al., 2019) of the original problem are global min-max points and recover the correct norm of $\boldsymbol{a}_i^*, \forall i$. Notice for the source data distribution $\boldsymbol{x} = (x_1, x_2, \cdots x_d) \sim \mathcal{D}$ with activation $\phi$, the marginal distribution of each $x_i$ follows $\phi(\mathcal{N}(0, \|\boldsymbol{a}_i^*\|^2))$ and is determined by $\|\boldsymbol{a}_i^*\|$. Therefore we have learned the marginal distribution for each entry $i$. It remains to learn the joint distribution.

## 4 LEARNING THE JOINT DISTRIBUTION

In the previous section, we utilize a (rectified) linear discriminator, such that each coordinate $v_i$ interacts with the $i$-th random variable. With the (rectified) linear discriminator, WGAN learns the correct $\|\boldsymbol{a}_i\|$, for all $i$. However, since there's no interaction between different coordinates of the random vector, we do not expect to learn the joint distribution with a linear discriminator.

To proceed, a natural idea is to use a quadratic discriminator $D_V(\boldsymbol{x}) := \boldsymbol{x}^\top V \boldsymbol{x} = \langle \boldsymbol{x}\boldsymbol{x}^\top, V \rangle$ to enforce component interactions. Similar to the previous section, we study the regularized version:

$$\min_{A \in \mathbb{R}^{d \times k}} \max_{V \in \mathbb{R}^{d \times d}} \left\{ f_2(A, V) - \frac{1}{2} \|V\|_F^2 \right\}, \tag{3}$$

where
$$f_2(A, V) = \mathbb{E}_{\boldsymbol{x}\sim\mathcal{D}} D_V(\boldsymbol{x}) - \mathbb{E}_{\boldsymbol{z}\sim\mathcal{N}(0, I_{k\times k})} D_V(\phi(A\boldsymbol{z}))$$
$$= \left\langle \mathbb{E}_{\boldsymbol{x}\sim\mathcal{D}} \left[ \boldsymbol{x}\boldsymbol{x}^\top \right] - \mathbb{E}_{\boldsymbol{z}\sim\mathcal{N}(0, I_{k\times k})} \left[ \phi(A\boldsymbol{z})\phi(A\boldsymbol{z})^\top \right], V \right\rangle.$$

By adding a regularizer on $V$ and explicitly maximizing over $V$:

$$g(A) \equiv \max_V \left\{ f_2(A, V) - \frac{1}{2}\|V\|_F^2 \right\}$$
$$= \frac{1}{2} \left\| \mathbb{E}_{\boldsymbol{x}\sim\mathcal{D}} \left[ \boldsymbol{x}\boldsymbol{x}^\top \right] - \mathbb{E}_{\boldsymbol{z}\sim\mathcal{N}(0, I_{k\times k})} \left[ \phi(A\boldsymbol{z})\phi(A\boldsymbol{z})^\top \right] \right\|_F^2.$$

In the next subsection, we first focus on analyzing the second-order stationary points of $g$, then we establish that gradient descent ascent converges to second-order stationary points of $g$.

### 4.1 GLOBAL CONVERGENCE FOR OPTIMIZING THE GENERATING PARAMETERS

We first assume that both $A$ and $A^*$ have unit row vectors, and then extend to general case since we already know how to learn the row norms from Section 3. To explicitly compute $g(A)$, we rely on the property of Hermite polynomials. Since normalized Hermite polynomials $\{h_i\}_{i=0}^\infty$ forms an orthonomal basis in the functional space, we rewrite the activation function as $\phi(\boldsymbol{x}) = \sum_{i=0}^\infty \sigma_i h_i$, where $\sigma_i$ is the $i$-th Hermite coefficient. We use the following claim:

**Claim 1** ((Ge et al., 2017) Claim 4.2). *Let $\phi$ be a function from $\mathbb{R}$ to $\mathbb{R}$ such that $\phi \in L^2(\mathbb{R}, e^{-x^2/2})$, and let its Hermite expansion be $\phi = \sum_{i=1}^\infty \sigma_i h_i$. Then, for any unit vectors $\boldsymbol{u}, \boldsymbol{v} \in \mathbb{R}^d$, we have that*

$$\mathbb{E}_{\boldsymbol{x}\sim\mathcal{N}(0, I_{d\times d})} \left[ \phi(\boldsymbol{u}^\top \boldsymbol{x})\phi(\boldsymbol{v}^\top \boldsymbol{x}) \right] = \sum_{i=0}^\infty \sigma_i^2 (\boldsymbol{u}^\top \boldsymbol{v})^i.$$

Therefore we could compute the value of $f_2$ explicitly using the Hermite polynomial expansion:

$$f_2(A, V) = \left\langle \sum_{i=0}^\infty \sigma_i^2 \left( (A^*(A^*)^\top)^{\circ i} - (AA^\top)^{\circ i} \right), V \right\rangle.$$

Here $X^{\circ i}$ is the Hadamard power operation where $(X^{\circ i})_{jk} = (X_{jk})^i$. Therefore we have:

$$g(A) = \frac{1}{2} \left\| \sum_{i=0}^\infty \sigma_i^2 \left( (A^*(A^*)^\top)^{\circ i} - (AA^\top)^{\circ i} \right) \right\|_F^2$$

We reparametrize with $Z = AA^\top$ and define $\tilde{g}(Z) = g(A)$ with individual component functions $\tilde{g}_{jk}(z) \equiv \frac{1}{2}(\sum_{i=0}^\infty \sigma_i^2((z_{jk}^*)^i - z^i))^2$. Accordingly $z_{jk}^* = \langle \boldsymbol{a}_j^*, \boldsymbol{a}_k^* \rangle$ is the $(j, k)$-th component of the ground truth covariance matrix $A^*(A^*)^\top$.

**Assumption 2.** *The activation function $\phi$ is an odd function plus constant. In other words, its Hermite expansion $\phi = \sum_{i=0}^\infty \sigma_i h_i$ satisfies $\sigma_i = 0$ for even $i \geq 2$. Additionally we assume $\sigma_1 \neq 0$.*

**Remark 2.** *Common activations like tanh and sigmoid satisfy Assumption 2.*

**Lemma 2.** *For activations including leaky ReLU and functions satisfying Assumption 2, $\tilde{g}(Z)$ has a unique stationary point where $Z = A^*(A^*)^\top$.*

Notice $\tilde{g}(Z) = \sum_{jk} \tilde{g}_{jk}(z_{jk})$ is separable across $z_{jk}$, where each $\tilde{g}_{jk}$ is a polynomial scalar function. Lemma 2 comes from the fact that the only zero point for $\tilde{g}_{jk}'$ is $z_{jk} = z_{jk}^*$, for odd activation $\phi$ and leaky ReLU. Then we migrate this good property to the original problem we want to solve:

**Problem 1.** *We optimize over function $g$ when $\|\boldsymbol{a}_i^*\| = 1, \forall i$:*

$$\min_A \left\{ g(A) = \frac{1}{2} \left\| \sum_{i=0}^\infty \sigma_i^2 \left( (A^*(A^*)^\top)^{\circ i} - (AA^\top)^{\circ i} \right) \right\|_F^2 \right\}$$
$$s.t. \quad \boldsymbol{a}_i^\top \boldsymbol{a}_i = 1, \forall i.$$

Existing work Journée et al. (2008) connects $\tilde{g}(Z)$ to the optimization over factorized version for $g(A)$ ($g(A) \equiv \tilde{g}(AA^\top)$). Specifically, when $k = d$, all second-order stationary points for $g(A)$ are first-order stationary points for $\tilde{g}(Z)$. Though $\tilde{g}$ is not convex, we are able to show that its first-order stationary points are global optima when the generator is sufficiently expressive, i.e., $k \geq k_0$. In reality we won't know the latent dimension $k_0$, therefore we just choose $k = d$ for simplicity. We make the following conclusion:

**Theorem 2.** *For activations including leaky ReLU and functions satisfying Assumption 2, when $k = d$, all second-order KKT points for problem 1 are its global minimum. Therefore alternating projected gradient descent-ascent on Eqn. (3) converges to $A : AA^\top = A^*(A^*)^\top$.*

The extension for non-unit vectors is straightforward, and we defer the analysis to the Appendix.

## 5  FINITE SAMPLE ANALYSIS

---

**Algorithm 1** Online stochastic gradient descent ascent on WGAN

---

1: **Input:** $n$ training samples: $\boldsymbol{x}_1, \boldsymbol{x}_2, \cdots \boldsymbol{x}_n$, where each $\boldsymbol{x}_i \sim \phi(A^*\boldsymbol{z}), \boldsymbol{z} \sim \mathcal{N}(0, I_{k \times k})$, learning rate for generating parameters $\eta$, number of iterations $T$.
2: Random initialize generating matrix $A^{(0)}$.
3: **for** $t = 1, 2, \cdots, T$ **do**
4:    Generate $m$ latent variables $\boldsymbol{z}_1^{(t)}, \boldsymbol{z}_2^{(t)}, \cdots, \boldsymbol{z}_m^{(t)} \sim \mathcal{N}(0, I_{k \times k})$ for the generator. The empirical function becomes

$$\tilde{f}_{m,n}^{(t)}(A, V) = \left\langle \frac{1}{m} \sum_{i=1}^m \phi(A\boldsymbol{z}_i^{(t)})\phi(A\boldsymbol{z}_i^{(t)})^\top - \frac{1}{n} \sum_{i=1}^n \boldsymbol{x}_i \boldsymbol{x}_i^\top, V \right\rangle - \frac{1}{2}\|V\|^2$$

5:    Gradient ascent on $V$ with optimal step-size $\eta_V = 1$:

$$V^{(t)} \leftarrow V^{(t)} - \eta_V \nabla_V \tilde{f}_{m,n}^{(t)}(A^{(t-1)}, V^{(t-1)}).$$

6:    Sample noise $\boldsymbol{e}$ uniformly from unit sphere
7:    Projected Gradient Descent on $A$, with constraints $C = \{A | (AA^\top)_{ii} = (A^*A^{*\top})_{ii}\}$ :

$$A^{(t)} \leftarrow \mathrm{Proj}_C(A^{(t-1)} - \eta(\nabla_A \tilde{f}_{m,n}^{(t)}(A^{(t-1)}, V^{(t)}) + \boldsymbol{e})).$$

8: **end for**
9: **Output:** $A^{(T)}(A^{(T)})^\top$

---

In this section, we consider analyzing Algorithm 1, i.e., gradient descent ascent on the following:

$$\tilde{f}_{m,n}^{(t)}(A, V) = \left\langle \frac{1}{m} \sum_{i=1}^m \phi(A\boldsymbol{z}_i^{(t)})\phi(A\boldsymbol{z}_i^{(t)})^\top - \frac{1}{n} \sum_{i=1}^n \boldsymbol{x}_i \boldsymbol{x}_i^\top, V \right\rangle - \frac{1}{2}\|V\|^2. \tag{4}$$

Notice in each iteration, gradient ascent with step-size 1 finds the optimal solution for $V$. By Danskin's theorem (Danskin, 2012), our min-max optimization is essentially gradient descent over $\tilde{g}_{m,n}^{(t)}(A) \equiv \max_V \tilde{f}_{m,n}^{(t)}(A, V) = \frac{1}{2}\|\frac{1}{m}\sum_{i=1}^m \phi(A\boldsymbol{z}_i^{(t)})\phi(A\boldsymbol{z}_i^{(t)})^\top - \frac{1}{n}\sum_{i=1}^n \boldsymbol{x}_i \boldsymbol{x}_i^\top\|_F^2$ with a batch of samples $\{\boldsymbol{z}_i^{(t)}\}$, i.e., stochastic gradient descent for $f_n(A) \equiv \mathbb{E}_{\boldsymbol{z}_i \sim \mathcal{N}(0, I_{k \times k}), \forall i \in [m]}[\tilde{g}_{m,n}(A)]$.

Therefore to bound the difference between $f_n(A)$ and the population risk $g(A)$, we analyze the sample complexity required on the observation side ($\boldsymbol{x}_i \sim \mathcal{D}, i \in [n]$) and the mini-batch size required on the learning part ($\phi(A\boldsymbol{z}_j), \boldsymbol{z}_j \sim \mathcal{N}(0, I_{k \times k}), j \in [m]$). We will show that with large enough $n, m$, the algorithm specified in Algorithm 1 that optimizes over the empirical risk will yield the ground truth covariance matrix with high probability.

Our proof sketch is roughly as follows:

1. With high probability, projected stochastic gradient descent finds a second order stationary point $\hat{A}$ of $f_n(\cdot)$ as shown in Theorem 31 of (Ge et al., 2015).

2. For sufficiently large $m$, our empirical objective, though a biased estimator of the population risk $g(\cdot)$, achieves good $\epsilon$-approximation to the population risk on both the gradient and Hessian (Lemmas 4&5). Therefore $\hat{A}$ is also an $\mathcal{O}(\epsilon)$-approximate second order stationary point (SOSP) for the population risk $g(A)$.

3. We show that any $\epsilon$-SOSP $\hat{A}$ for $g(A)$ yields an $\mathcal{O}(\epsilon)$-first order stationary point (FOSP) $\hat{Z} \equiv \hat{A}\hat{A}^\top$ for the semi-definite programming on $\tilde{g}(Z)$ (Lemma 6).

4. We show that any $\mathcal{O}(\epsilon)$-FOSP of function $\tilde{g}(Z)$ induces at most $\mathcal{O}(\epsilon)$ absolute error compared to the ground truth covariance matrix $Z^* = A^*(A^*)^\top$ (Lemma 7).

## 5.1 OBSERVATION SAMPLE COMPLEXITY

For simplicity, we assume the activation and its gradient satisfy Lipschitz continuous, and let the Lipschitz constants be 1 w.l.o.g.:

**Assumption 3.** *Assume the activation is* 1*-Lipschitz and* 1*-smooth.*

To estimate observation sample complexity, we will bound the gradient and Hessian for the population risk and empirical risk on the observation samples:

$$g(A) \equiv \frac{1}{2} \left\| \mathbb{E}_{\boldsymbol{x} \sim \mathcal{D}} \left[ \boldsymbol{x}\boldsymbol{x}^\top \right] - \mathbb{E}_{\boldsymbol{z} \sim \mathcal{N}(0, I_{k \times k})} \left[ \phi(A\boldsymbol{z})\phi(A\boldsymbol{z})^\top \right] \right\|_F^2, \text{ and}$$

$$g_n(A) \equiv \frac{1}{2} \left\| \frac{1}{n} \sum_{i=1}^n \boldsymbol{x}_i \boldsymbol{x}_i^\top - \mathbb{E}_{\boldsymbol{z} \sim \mathcal{N}(0, I_{k \times k})} \left[ \phi(A\boldsymbol{z})\phi(A\boldsymbol{z})^\top \right] \right\|_F^2.$$

**Claim 2.**
$$\nabla g(A) - \nabla g_n(A) = 2\mathbb{E}_{\boldsymbol{z}} \left[ diag(\phi'(A\boldsymbol{z}))(X - X_n)\phi(A\boldsymbol{z})\boldsymbol{z}^\top \right],$$
*where* $X = \mathbb{E}_{\boldsymbol{x} \sim \mathcal{D}}[\boldsymbol{x}\boldsymbol{x}^\top]$, *and* $X_n = \frac{1}{n}\sum_{i=1}^n \boldsymbol{x}_i\boldsymbol{x}_i^\top$. *The directional derivative with arbitrary direction* $B$ *is:*

$$D\nabla g(A)[B] - D\nabla g_n(A)[B] = 2\mathbb{E}_{\boldsymbol{z}} \left[ diag(\phi'(A\boldsymbol{z}))(X_n - X)\phi'(A\boldsymbol{z}) \circ (B\boldsymbol{z})\boldsymbol{z}^\top \right]$$
$$+ 2\mathbb{E}_{\boldsymbol{z}} \left[ diag(\phi''(A\boldsymbol{z}) \circ (B\boldsymbol{z}))(X_n - X)\phi(A\boldsymbol{z})\boldsymbol{z}^\top \right]$$

**Lemma 3.** *Suppose the activation satisfies Assumption 3.* $\Pr[\|X - X_n\| \leq \epsilon \|X\|] \geq 1 - \delta$, *for* $n \geq \tilde{\Theta}(d/\epsilon^2 \log^2(1/\delta))^2$.

**Lemma 4.** *Suppose the activation satisfies Assumption 2&3. With samples* $n \geq \tilde{\Theta}(d/\epsilon^2 \log^2(1/\delta))$, $\|\nabla g(A) - \nabla g_n(A)\|_2 \leq \mathcal{O}(\epsilon d\|A\|_2)$ *with probability* $1 - \delta$. *Meanwhile,* $\|D\nabla g(A)[B] - D\nabla g_n(A)[B]\|_2 \leq \mathcal{O}(\epsilon d^{3/2}\|A\|_2\|B\|_2)$ *with probability* $1 - \delta$.

## 5.2 BOUNDING MINI-BATCH SIZE

Normally for empirical risk for supervised learning, the mini-batch size can be arbitrarily small since the estimator of the gradient is unbiased. However in the WGAN setting, notice for each iteration, we randomly sample a batch of random variables $\{\boldsymbol{z}_i\}_{i \in [m]}$, and obtain a gradient of $\tilde{g}_{m,n}(A) \equiv \frac{1}{2} \left\| \frac{1}{n}\sum_{i=1}^n \boldsymbol{x}_i\boldsymbol{x}_i^\top - \frac{1}{m}\sum_{j=1}^m \phi(A\boldsymbol{z}_j)\phi(A\boldsymbol{z}_j)^\top \right\|_F^2$, in Algorithm 1. However, the finite sum is inside the Frobenius norm and the gradient on each mini-batch may no longer be an unbiased estimator for our target $g_n(A) = \frac{1}{2} \left\| \frac{1}{n}\sum_{i=1}^n \boldsymbol{x}_i\boldsymbol{x}_i^\top - \mathbb{E}_{\boldsymbol{z}}\left[\phi(A\boldsymbol{z})\phi(A\boldsymbol{z})^\top\right] \right\|_F^2$.

In other words, we conduct stochastic gradient descent over the function $f(A) \equiv \mathbb{E}_{\boldsymbol{z}}\tilde{g}_{m,n}(A)$. Therefore we just need to analyze the gradient error between this $f(A)$ and $g_n(A)$ (i.e. $\tilde{g}_{m,n}$ is almost an unbiased estimator of $g_n$). Finally with the concentration bound derived in last section, we get the error bound between $f(A)$ and $g(A)$.

**Lemma 5.** *The empirical risk* $\tilde{g}_{m,n}$ *is almost an unbiased estimator of* $g_n$. *Specifically, the expected function* $f(A) = \mathbb{E}_{\boldsymbol{z}_i \sim \mathcal{N}(0, I_{k \times k}), i \in [m]}[\tilde{g}_{m,n}]$ *satisfies:*

$$\|\nabla f(A) - \nabla g_n(A)\| \leq \mathcal{O}(\frac{1}{m}\|A\|^3 d^2).$$

---

[2]$\tilde{\Theta}$ hides log factors of $d$ for simplicity.

*For arbitrary direction matrix B,*

$$\|D\nabla f(A)[B] - D\nabla g_n(A)[B]\| \leq \mathcal{O}(\frac{1}{m}\|B\|\|A\|^3 d^{5/2}).$$

In summary, we conduct concentration bound over the observation samples and mini-batch sizes, and show the gradient of $f(A)$ that Algorithm 1 is optimizing over has close gradient and Hessian with the population risk $g(A)$. Therefore a second-order stationary point (SOSP) for $f(A)$ (that our algorithm is guaranteed to achieve) is also an $\epsilon$ approximated SOSP for $g(A)$. Next we show such a point also yield an $\epsilon$ approximated first-order stationary point of the reparametrized function $\tilde{g}(Z) \equiv g(A), \forall Z = AA^\top$.

## 5.3 RELATION ON APPROXIMATE OPTIMALITY

In this section, we establish the relationship between $\tilde{g}$ and $g$. We present the general form of our target Problem 1:

$$\min_{A \in \mathbb{R}^{d \times k}} \quad g(A) \equiv \tilde{g}(AA^\top) \tag{5}$$
$$\text{s.t.} \quad \text{Tr}(A^\top X_i A) = y_i, X_i \in \mathbb{S}, y_i \in \mathbb{R}, i = 1, \cdots, n.$$

Similar to the previous section, the stationary property might not be obvious on the original problem. Instead, we could look at the re-parametrized version as:

$$\min_{Z \in \mathbb{S}} \quad \tilde{g}(Z) \tag{6}$$
$$\text{s.t.} \quad \text{Tr}(X_i Z) = y_i, X_i \in \mathbb{S}, y_i \in \mathbb{R}, i = 1, \cdots, n,$$
$$Z \succeq 0,$$

**Definition 1.** *A matrix $A \in \mathbb{R}^{d \times k}$ is called an $\epsilon$-approximate second-order stationary point ($\epsilon$-SOSP) of Eqn. (5) if there exists a vector $\lambda$ such that:*

$$\begin{cases} \text{Tr}(A^\top X_i A) = y_i, i \in [n] \\ \|(\nabla_Z \tilde{g}(AA^\top) - \sum_{i=1}^n \lambda_i X_i)\tilde{\boldsymbol{a}}_j\| \leq \epsilon\|\tilde{\boldsymbol{a}}_j\|, \quad \{\tilde{\boldsymbol{a}}_j\}_j \text{ span the column space of } A \\ \text{Tr}(B^\top D\nabla_A \mathcal{L}(A, \lambda)[B]) \geq -\epsilon\|B\|^2, \quad \forall B \text{ s.t. } \text{Tr}(B^\top X_i A) = 0 \end{cases}$$

*Here $\mathcal{L}(A, \lambda)$ is the Lagrangian form $\tilde{g}(AA^\top) - \sum_{i=1}^n \lambda_i(\text{Tr}(A^\top X_i A) - y_i)$.*

Specifically, when $\epsilon = 0$ the above definition is exactly the second-order KKT condition for optimizing (5). Next we present the approximate first-order KKT condition for (6):

**Definition 2.** *A symmetric matrix $Z \in \mathbb{S}^n$ is an $\epsilon$-approximate first order stationary point of function (6) ($\epsilon$-FOSP) if and only if there exist a vector $\sigma \in \mathbb{R}^m$ and a symmetric matrix $S \in \mathbb{S}$ such that the following holds:*

$$\begin{cases} \text{Tr}(X_i Z) = y_i, i \in [n] \\ Z \succeq 0, \\ S \succeq -\epsilon I, \\ \|S\tilde{\boldsymbol{a}}_j\| \leq \epsilon\|\tilde{\boldsymbol{a}}_j\|, \quad \{\tilde{\boldsymbol{a}}_j\}_j \text{ span the column space of } Z \\ S = \nabla_Z \tilde{g}(Z) - \sum_{i=1}^n \sigma_i X_i. \end{cases}$$

**Lemma 6.** *Let latent dimension $k = d$. For an $\epsilon$-SOSP of function (5) with $A$ and $\lambda$, it infers an $\epsilon$-FOSP of function (6) with $Z, \sigma$ and $S$ that satisfies: $Z = AA^\top, \sigma = \lambda$ and $S = \nabla_Z \tilde{g}(AA^\top) - \sum_i \lambda_i X_i$.*

Now it remains to show an $\epsilon$-FOSP of $\tilde{g}(Z)$ indeed yields a good approximation for the ground truth parameter matrix.

**Lemma 7.** *If $Z$ is an $\epsilon$-FOSP of function (6), then $\|Z - Z^*\|_F \leq \mathcal{O}(\epsilon)$. Here $Z^* = A^*(A^*)^\top$ is the optimal solution for function (6).*

Together with the previous arguments, we finally achieve our main theorem on connecting the recovery guarantees with the sample complexity and batch size[3]:

**Theorem 3.** *For arbitrary $\delta < 1, \epsilon$, given small enough learning rate $\eta < 1/poly(d, 1/\epsilon, \log(1/\delta))$, let sample size $n \geq \tilde{\Theta}(d^5/\epsilon^2 \log^2(1/\delta))$, batch size $m \geq \mathcal{O}(d^5/\epsilon)$, for large enough $T = poly(1/\eta, 1/\epsilon, d, \log(1/\delta))$, the output of Algorithm 1 satisfies $\|A^{(T)}(A^{(T)})^\top - Z^*\|_F \leq \mathcal{O}(\epsilon)$ with probability $1 - \delta$, under Assumptions 2 & 3 and $k = d$.*

---

[3] The exact error bound comes from the fact that when diagonal terms of $AA^\top$ are fixed, $\|A\|_2 = \mathcal{O}(\sqrt{d})$.

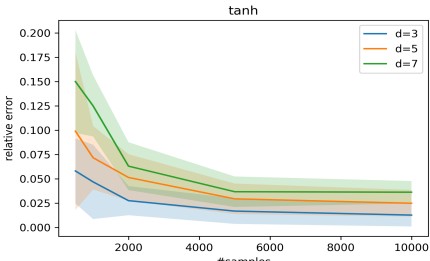 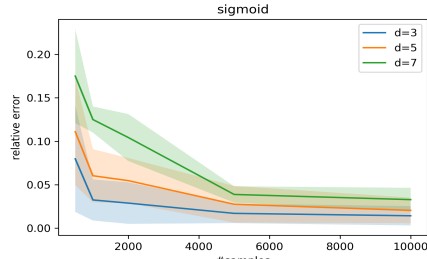

Figure 1: Recovery error ($\|AA^\top - Z^*\|_F$) with different observed sample sizes $n$ and output dimension $d$.

## 6 SIMULATIONS

In this section, we provide simple experimental results to validate the performance of stochastic gradient descent ascent and provide experimental support for our theory.

We focus on Algorithm 1 that targets to recover the parameter matrix. We conduct a thorough empirical studies on three joint factors that might affect the performance: the number of observed samples $m$ (we set $n = m$ as in general GAN training algorithms), the different choices of activation function $\phi$, and the output dimension $d$. In Figure 1 we plot the relative error for parameter estimation decrease over the increasing sample complexity. We fix the hidden dimension $k = 2$, and vary the output dimension over $\{3, 5, 7\}$ and sample complexity over $\{500, 1000, 2000, 5000, 10000\}$. Reported values are averaged from 20 runs and we show the standard deviation with the corresponding colored shadow. Clearly the recovery error decreases with higher sample complexity and smaller output dimension.

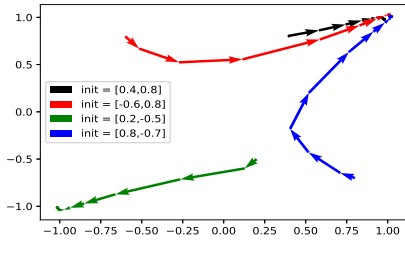 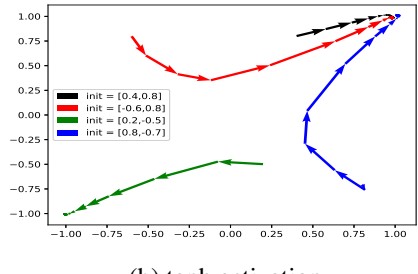

(a) leaky ReLU activation ($\alpha = 0.2$)          (b) tanh activation

Figure 2: Comparisons of different performance with leakyReLU and tanh activations. Same color starts from the same starting point. For both cases, parameters always converge to true covariance matrix. Each arrow indicates the progress of 500 iteration steps.

To visually demonstrate the learning process, we also include a simple comparison for different $\phi$: i.e. leaky ReLU and tanh activations, when $k = 1$ and $d = 2$. We set the ground truth covariance matrix to be $[1, 1; 1, 1]$, and therefore a valid result should be $[1, 1]$ or $[-1, -1]$. From Figure 2 we could see that for both leaky ReLU and tanh, the stochastic gradient descent ascent performs similarly with exact recovery of the ground truth parameters.

## 7 CONCLUSION

We analyze the convergence of stochastic gradient descent ascent for Wasserstein GAN on learning a single layer generator network. We show that stochastic gradient descent ascent algorithm attains the global min-max point, and provably recovers the parameters of the network with $\epsilon$ absolute error measured in Frobenius norm, from $\tilde{\Theta}(d^5/\epsilon^2)$ i.i.d samples.

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

# A    OMITTED PROOF FOR LEARNING THE DISTRIBUTION

## A.1    STATIONARY POINT FOR MATCHING FIRST MOMENT

*Proof of Lemma 1.* To start with, we consider odd-plus-constant monotone increasing activations. Notice that by proposing a rectified linear discriminator, we have essentially modified the activation function as $\tilde{\phi} := R(\phi - C)$, where $C = \frac{1}{2}(\phi(x) + \phi(-x))$ is the constant bias term of $\phi$. Observe that we can rewrite the objective $\bar{f}_1$ for this case as follows:

$$f_1(A, \boldsymbol{v}) = \mathbb{E}_{\boldsymbol{z} \sim \mathcal{N}(0, I_{k_0 \times k_0})} \boldsymbol{v}^\top \tilde{\phi}(A^* \boldsymbol{z}) - \mathbb{E}_{\boldsymbol{z} \sim \mathcal{N}(0, I_{k \times k})} \boldsymbol{v}^\top \tilde{\phi}(A \boldsymbol{z}).$$

Moreover, notice that $\tilde{\phi}$ is positive and increasing on its support which is $[0, +\infty)$.

Now let us consider the other case in our statement where $\phi$ has a positive and monotone increasing even component in $[0, +\infty)$. In this case, let us take:

$$\tilde{\phi}(x) = \begin{cases} \phi(x) + \phi(-x), & x \geq 0 \\ 0, & \text{o.w.} \end{cases}$$

Because of the symmetry of the Gaussian distribution, we can rewrite the objective function for this case as follows:

$$f_1(A, \boldsymbol{v}) = \mathbb{E}_{\boldsymbol{z} \sim \mathcal{N}(0, I_{k_0 \times k_0})} \boldsymbol{v}^\top \tilde{\phi}(A^* \boldsymbol{z}) - \mathbb{E}_{\boldsymbol{z} \sim \mathcal{N}(0, I_{k \times k})} \boldsymbol{v}^\top \tilde{\phi}(A \boldsymbol{z}).$$

Moreover, notice that $\tilde{\phi}$ is positive and increasing on its support which is $[0, +\infty)$.

To conclude, in both cases, the optimization objective can be written as follows, where $\tilde{\phi}$ satisfies Assumption 1.2 and is only non-zero on $[0, +\infty)$.

$$f_1(A, \boldsymbol{v}) = \mathbb{E}_{\boldsymbol{z} \sim \mathcal{N}(0, I_{k_0 \times k_0})} \boldsymbol{v}^\top \tilde{\phi}(A^* \boldsymbol{z}) - \mathbb{E}_{\boldsymbol{z} \sim \mathcal{N}(0, I_{k \times k})} \boldsymbol{v}^\top \tilde{\phi}(A \boldsymbol{z}).$$

The stationary points of the above objective satisfy:

$$\begin{cases} \nabla_{\boldsymbol{v}} f_1(A, \boldsymbol{v}) = \mathbb{E}_{\boldsymbol{z} \sim \mathcal{N}(0, I_{k_0 \times k_0})} \tilde{\phi}(A^* \boldsymbol{z}) - \mathbb{E}_{\boldsymbol{z} \sim \mathcal{N}(0, I_{k \times k})} \tilde{\phi}(A \boldsymbol{z}) = 0, \\ \nabla_{\boldsymbol{a}_j} f_1(A, \boldsymbol{v}) = -\mathbb{E}_{\boldsymbol{z} \sim \mathcal{N}(0, I_{k \times k})} v_j \tilde{\phi}'(\boldsymbol{a}_j^\top \boldsymbol{z}) \boldsymbol{z} = 0. \end{cases}$$

We focus on the gradient over $\boldsymbol{v}$. To achieve $\nabla_{\boldsymbol{v}} f_1(A, \boldsymbol{v}) = 0$, the stationary point satisfies:

$$\forall j, \mathbb{E}_{\boldsymbol{z} \sim \mathcal{N}(0, I_{k_0 \times k_0})} \tilde{\phi}((\boldsymbol{a}_j^*)^\top \boldsymbol{z}) = \mathbb{E}_{\boldsymbol{z} \sim \mathcal{N}(0, I_{k \times k})} \tilde{\phi}(\boldsymbol{a}_j^\top \boldsymbol{z}), \text{ i.e.}$$

$$\forall j, \mathbb{E}_{x \sim \mathcal{N}(0, \|\boldsymbol{a}_j^*\|^2)} \tilde{\phi}(x) = \mathbb{E}_{x' \sim \mathcal{N}(0, \|\boldsymbol{a}_j\|^2)} \tilde{\phi}(x'). \tag{7}$$

To recap, for activations $\phi$ that follow Assumption 1, in both cases we have written the necessary condition on stationary point to be Eqn. (7), where $\tilde{\phi}$ is defined differently for odd or non-odd activations, but in both cases it is positive and monotone increasing on its support $[0, \infty)$. We then argue the only solution for Eqn. (7) satisfies $\|\boldsymbol{a}_j\| = \|\boldsymbol{a}_j^*\|, \forall j$. This follows directly from the following claim:

**Claim 3.** *The function $h(\alpha) := \mathbb{E}_{x \sim \mathcal{N}(0, \alpha^2)} f(x), \alpha > 0$ is a monotone increasing function if $f$ is positive and monotone increasing on its support $[0, \infty)$.*

We could see from Claim 3 that the LHS and RHS of Eqn. (7) is simply $h(\|\boldsymbol{a}_j\|)$ and $h(\|\boldsymbol{a}_j^*\|)$ for each $j$. Now that $h$ is an monotone increasing function, the unique solution for $h(\|\boldsymbol{a}_j\|) = h(\|\boldsymbol{a}_j^*\|)$ is to match the norm: $\|\boldsymbol{a}_j\| = \|\boldsymbol{a}_j^*\|, \forall j$.

*Proof of Claim 3.*

$$\begin{aligned} h(\alpha) &= \mathbb{E}_{x \sim \mathcal{N}(0, \alpha^2)} f(x) \\ &= \int_0^\infty f(x) e^{-\frac{x^2}{2\alpha^2}} dx \\ &\stackrel{y := x/\alpha}{=} \int_0^\infty \alpha f(\alpha y) e^{-\frac{y^2}{2}} dy \\ &= \mathbb{E}_{y \sim \mathcal{N}(0,1)} \alpha f(\alpha y). \end{aligned}$$

Notice $h'(\alpha) = \mathbb{E}_{x \sim \mathcal{N}(0,1)}[\alpha x f'(\alpha x) + f(\alpha x)]$. Since $f$, $f'$, and $\alpha > 0$, and we only care about the support of $f$ where $x$ is also positive, therefore $h'$ is always positive and $h$ is monotone increasing. $\square$

To sum up, at stationary point where $\nabla f_1(A, \boldsymbol{v}) = 0$, we have

$$\forall i, \|\boldsymbol{a}_i^*\| = \|\boldsymbol{a}_i\|.$$

$\square$

## A.2   Proof of Theorem 1

*Proof of Theorem 1.* We will take optimal gradient ascent steps with learning rate 1 on the discriminator side $\boldsymbol{v}$, hence the function we will actually be optimizing over becomes (using the notation for $\tilde{\phi}$ from section A.1):

$$h(A) = \max_{\boldsymbol{v}} f_1(A, \boldsymbol{v}) = \frac{1}{2} \left\| \mathbb{E}_{\boldsymbol{z} \sim \mathcal{N}(0, I_{k_0 \times k_0})} \tilde{\phi}(A^* \boldsymbol{z}) - \mathbb{E}_{\boldsymbol{z} \sim \mathcal{N}(0, I_{k \times k})} \tilde{\phi}(A \boldsymbol{z}) \right\|^2.$$

We just want to verify that there's no spurious local minimum for $h(A)$. Notice there's no interaction between each row vector of $A$. Therefore we instead look at each $h_i := \frac{1}{2} \left( \mathbb{E}_{\boldsymbol{z} \sim \mathcal{N}(0, I_{k_0 \times k_0})} \tilde{\phi}((\boldsymbol{a}_i^*)^\top \boldsymbol{z}) - \mathbb{E}_{\boldsymbol{z} \sim \mathcal{N}(0, I_{k \times k})} \tilde{\phi}(\boldsymbol{a}_i^\top \boldsymbol{z}) \right)^2$ for each $i$. Now $\nabla h_i(\boldsymbol{a}_i) = -\left( \mathbb{E}_{\boldsymbol{z} \sim \mathcal{N}(0, I_{k_0 \times k_0})} \tilde{\phi}((\boldsymbol{a}_i^*)^\top \boldsymbol{z}) - \mathbb{E}_{\boldsymbol{z} \sim \mathcal{N}(0, I_{k \times k})} \tilde{\phi}(\boldsymbol{a}_i^\top \boldsymbol{z}) \right) (\mathbb{E}_{\boldsymbol{z} \sim \mathcal{N}(0, I_{k \times k})} \boldsymbol{z} \tilde{\phi}'(\boldsymbol{a}_i^\top \boldsymbol{z}))$. Due to the symmetry of the Gaussian, we take $\boldsymbol{a}_i = a \boldsymbol{e}_1$, where $a = \|\boldsymbol{a}_i\|$. It is easy to see that checking whether $\mathbb{E}_{\boldsymbol{z} \sim \mathcal{N}(0, I_{k \times k})} \boldsymbol{z} \tilde{\phi}'(\boldsymbol{a}_i^\top \boldsymbol{z}) = 0$ is equivalent to checking whether $\mathbb{E}_{z_1 \sim \mathcal{N}(0,1)} z_1 \tilde{\phi}'(a z_1) = 0$.

Recall that $\tilde{\phi}$ is supported on $[0, +\infty)$ and it is monotonically increasing on its support. Hence, $\mathbb{E}_{z_1 \sim \mathcal{N}(0,1)} z_1 \tilde{\phi}'(a z_1) \neq 0$ unless $a = 0$. Hence, suppose $\|\boldsymbol{a}_i\| \neq 0, \forall i$. Then $\nabla_A h(A) = 0$ iff $h(A) = 0$, i.e. $\mathbb{E}_{\boldsymbol{z} \sim \mathcal{N}(0, I_{k_0 \times k_0})} \tilde{\phi}(A^* \boldsymbol{z}) = \mathbb{E}_{\boldsymbol{z} \sim \mathcal{N}(0, I_{k \times k})} \tilde{\phi}(A \boldsymbol{z})$.

Therefore all stationary points of $h(A)$ are global minima where $\mathbb{E}_{\boldsymbol{z} \sim \mathcal{N}(0, I_{k_0 \times k_0})} \tilde{\phi}(A^* \boldsymbol{z}) = \mathbb{E}_{\boldsymbol{z} \sim \mathcal{N}(0, I_{k \times k})} \tilde{\phi}(A \boldsymbol{z})$ and according to Lemma 1, this only happens when $\|\boldsymbol{a}_i\| = \|\boldsymbol{a}_i^*\|, \forall i \in [d]$. $\square$

## A.3   Stationary Points for WGAN with Quadratic Discriminator

*Proof of Lemma 2.* To study the stationary point for $\tilde{g}(Z) = \sum_{jk} \tilde{g}_{jk}(z_{jk})$, we look at individual $\tilde{g}_{jk}(z) \equiv \frac{1}{2} (\sum_{i=0}^{\infty} \sigma_i^2 ((z_{jk}^*)^i - z^i))^2$.

Notice for odd-plus-constant activations, $\sigma_i$ is zero for even $i > 0$. Recall our assumption in Lemma 2 also requires that $\sigma_1 \neq 0$. Since the analysis is invariance to the position of the matrix $Z$, we simplify the notation here and essentially want to study the stationary point for $f(a) = \frac{1}{2} (\sum_{i \text{ odd}} \sigma_i^2 (a^i - b^i))^2$ for some constant $b$ and $\sigma_i$, where $\sigma_1 \neq 0$[4].

$$
\begin{aligned}
f'(a) &= \left( \sum_{i \text{ odd}} \sigma_i^2 (a^i - b^i) \right) \left( \sum_{i \text{ odd}} i \sigma_i^2 a^{i-1} \right) \\
&= (a - b) \left( \sigma_1^2 + \sum_{i \geq 3 \text{ odd}} \sigma_i^2 \frac{a^i - b^i}{a - b} \right) \left( \sigma_1^2 + \sum_{i \geq 3 \text{ odd}} i \sigma_i^2 a^{i-1} \right) \\
&= (a - b)(\text{I})(\text{II}).
\end{aligned}
$$

Notice now $f'(a) = 0 \Leftrightarrow a = b$. This is because the polynomial $f'(a)$ is factorized to $a - b$ and two factors I and II that are always positive. Notice here we use $\frac{a^i - b^i}{a - b}$ to denote $\sum_{j=0}^{i} a^j b^{i-j}$, which is always nonnegative. This is simply because $a^i - b^i$ always shares the same sign as $a - b$ when $i$ is odd. Therefore I$= \sigma_1^2 + \sum_{i \geq 3 \text{ odd}} \sigma_i^2 \frac{a^i - b^i}{a - b} > 0, \forall a$.

---

[4]The zero component has been cancelled out.

Meanwhile, since $a^{i-1}$ is always nonnegative for each odd $i$, we have II$= \sigma_1^2 + \sum_{i \geq 3 \text{ odd}} i\sigma_i^2 a^{i-1}$ is also always positive for any $a$.

Next, for activation like ReLU, loss $\tilde{g}_{jk}(z) = \frac{1}{2}(h(z) - h(z_{jk}^*))^2$, where $h(x) = \frac{1}{\pi}(\sqrt{1 - x^2} + (\pi - \cos^{-1}(x))x)$ (Daniely et al., 2016). Therefore $h'(-1) = 0$ for any $z_{jk}^*$. This fact prevents us from getting the same conclusion for ReLU.

However, for leaky ReLU with coefficient of leakage $\alpha \in (0, 1)$, $\phi(x) = \max\{x, \alpha x\} = (1 - \alpha)\text{ReLU}(x) + \alpha x$.

We have

$$
\begin{aligned}
&\mathbb{E}_{\boldsymbol{z} \sim \mathcal{N}(0, I_{k \times k})} \left[ \phi(\boldsymbol{a}_i^\top \boldsymbol{z}) \phi(\boldsymbol{a}_j^\top \boldsymbol{z}) \right] \\
=&(1 - \alpha)^2 \mathbb{E}_{\boldsymbol{z}} \text{ReLU}(\boldsymbol{a}_i^\top \boldsymbol{z}) \text{ReLU}(\boldsymbol{a}_j^\top \boldsymbol{z}) + (1 - \alpha)\alpha \mathbb{E}_{\boldsymbol{z}} \text{ReLU}(\boldsymbol{a}_i^\top \boldsymbol{z}) \boldsymbol{a}_j^\top \boldsymbol{z} \\
&+ (1 - \alpha)\alpha \mathbb{E}_{\boldsymbol{z}} \boldsymbol{a}_i^\top \boldsymbol{z} \text{ReLU}(\boldsymbol{a}_j^\top \boldsymbol{z}) + \alpha^2 \mathbb{E}_{\boldsymbol{z}} \boldsymbol{a}_i^\top \boldsymbol{z} \boldsymbol{a}_j^\top \boldsymbol{z} \\
=&(1 - \alpha)^2 h(\boldsymbol{a}_i^\top \boldsymbol{a}_j) + \alpha \boldsymbol{a}_i^\top \boldsymbol{a}_j
\end{aligned}
$$

Therefore for leaky ReLU $\tilde{g}_{jk}(z) = \frac{1}{2}((1 - \alpha)^2(h(z) - h(z_{jk^*})) + \alpha(z - z_{jk}^*))^2$, and $\tilde{g}'_{jk}(z) = ((1 - \alpha)^2(h(z) - h(z_{jk^*})) + \alpha(z - z_{jk}^*))((1 - \alpha)^2 h'(z) + \alpha)$. Now with $\alpha > 0$, $(1 - \alpha)^2 h'(z) + \alpha \geq \alpha$ for all $z$ and $\tilde{g}_{jk}(z) = 0 \Leftrightarrow z = z_{jk}^*$.

To sum up, for odd activations and leaky ReLU, since each $\tilde{g}_{jk}(z)$ only has stationary point of $z = z_{jk}^*$, the stationary point $Z$ of $\tilde{g}(Z) = \sum_{jk} \tilde{g}_{jk}$ also satisfy $Z = Z^* = A^*(A^*)^\top$.

$\square$

*Proof of Theorem 2.* Instead of directly looking at the second-order stationary point of Problem 1, we look at the following problem on its reparametrized version:

**Problem 2.**

$$
\min_Z \left\{ \tilde{g}(Z) = \frac{1}{2} \left\| \sum_{i=0}^{\infty} \sigma_i^2 \left( (Z^*)^{\circ i} - Z^{\circ i} \right) \right\|_F^2 \right\}
$$

$$
\begin{aligned}
\textit{s.t.} \quad & z_{ii} = 1, \forall i. \\
& Z \succeq 0.
\end{aligned}
$$

*Here $Z^* = A^*(A^*)^\top$ and satisfies $z_{ii}^* = 1, \forall i$.*

Compared to function $g$ in the original problem 1, it satisfies that $\tilde{g}(AA^\top) \equiv g(A)$.

A matrix $Z$ satisfies the first-order stationary point for Problem 2 if there exists a vector $\sigma$ such that:

$$
\begin{cases}
z_{ii} = 1, \\
Z \succeq 0, \\
S \succeq 0, \\
SZ = 0, \\
S = \nabla_Z g(Z) - \text{diag}(\sigma).
\end{cases}
$$

Therefore for a stationary point $Z$, since $Z^* = A^*(A^*)^\top \succeq 0$, and $S \succeq 0$, we have $\langle S, Z^* - Z \rangle = \langle S, Z^* \rangle \geq 0$. Meanwhile,

$$
\begin{aligned}
&\langle Z^* - Z, S \rangle \\
=&\langle Z^* - Z, \nabla_Z f(Z) - \text{diag}(\sigma) \rangle \\
=&\langle Z^* - Z, \nabla_Z f(Z) \rangle \qquad (\text{diag}(Z^* - Z) = 0) \\
=&\sum_{i,j} (z_{ij}^* - z_{ij}) g_{ij}'(z_{ij}) \\
=&\sum_{i,j} (z_{ij} - z_{ij}^*) P(z_{ij})(z_{ij}^* - z_{ij}) \\
&\qquad\qquad (\text{Refer to proof of Lemma 2 for the value of } g') \\
=&-\sum_{ij} (z_{ij} - z_{ij}^*)^2 P(z_{ij}) \\
\leq&0 \qquad\qquad\qquad (P \text{ is always positive})
\end{aligned}
$$

Therefore $\langle S, Z^* - Z \rangle = 0$, and this only happens when $Z = Z^*$.

Finally, from Journée et al. (2008) we know that any first-order stationary point for Problem 2 is a second-order stationary point for our original problem 1 [5]. Therefore we conclude that all second-order stationary point for Problem 1 are global minimum $A$: $AA^\top = A^*(A^*)^\top$. □

### A.4 LANDSCAPE ANALYSIS FOR NON-UNIT GENERATING VECTORS

In the previous argument, we simply assume that the norm of each generating vectors $\boldsymbol{a}_i$ to be 1. This practice simplifies the computation but is not practical. Since we are able to estimate $\|\boldsymbol{a}_i\|$ for all $i$ first, we could analyze the landscape of our loss function for general matrix $A$.

The main tool is to use the multiplication theorem of Hermite functions:

$$
h_n^\alpha(x) := h_n(\alpha x) = \sum_{i=0}^{\lfloor \frac{n}{2} \rfloor} \alpha^{n-2i}(\alpha^2 - 1)^i \binom{n}{2i} \frac{(2i)!}{i!} 2^{-i} h_{n-2i}(x).
$$

For the ease of notation, we denote the coefficient as $\eta_\alpha^{n,i} := \alpha^{n-2i}(\alpha^2 - 1)^i \binom{n}{2i} \frac{(2i)!}{i!} 2^{-i}$. We extend the calculations for Hermite inner product for non-standard distributions.

**Lemma 8.** *Let $(x, y)$ be normal variables that follow joint distribution $\mathcal{N}(0, [[\alpha^2, \alpha\beta\rho]; [\alpha\beta\rho, \beta^2]])$. Then,*

$$
\mathbb{E}[h_m(x)h_n(y)] = \begin{cases} \sum_{i=0}^{\lfloor \frac{l}{2} \rfloor} \eta_\alpha^{l,i} \eta_\beta^{l,i} \rho^{l-2i} & \text{if } m \equiv n \ (mod \ 2) \\ 0 & \text{o.w.} \end{cases} \tag{8}
$$

*Here $l = \min\{m, n\}$.*

---

[5]Throughout the analysis for low rank optimization in Journée et al. (2008), they require function $\tilde{g}(Z)$ to be convex. However, by carefully scrutinizing the proof, one could see that this condition is not required in building the connection of first-order and second-order stationary points of $g(A)$ and $\tilde{g}(Z)$. For more cautious readers, we also show a relaxed version in the next section, where the equivalence of SOSP of $g$ and FOSP of $\tilde{g}$ is a special case of it.

*Proof.* Denote the normalized variables $\hat{x} = x/\alpha$, $\hat{y} = y/\beta$. Let $l = \min\{m, n\}$.

$$\mathbb{E}[h_m(x)h_n(y)]$$
$$=\mathbb{E}[h_m^\alpha(\hat{x})h_n^\beta(\hat{y})]$$
$$=\sum_{i=0}^{\lfloor \frac{m}{2} \rfloor}\sum_{j=0}^{\lfloor \frac{n}{2} \rfloor} \eta_\alpha^{m,i}\eta_\beta^{n,j}\mathbb{E}[h_{m-2i}(\hat{x})h_{n-2j}(\hat{y})]$$
$$=\sum_{i=0}^{\lfloor \frac{m}{2} \rfloor}\sum_{j=0}^{\lfloor \frac{n}{2} \rfloor} \eta_\alpha^{m,i}\eta_\beta^{n,j}\delta_{(m-2i),(n-2j)}\rho^{n-2j} \qquad \text{(Lemma ??)}$$
$$=\begin{cases} \sum_{i=0}^{\lfloor \frac{l}{2} \rfloor}\eta_\alpha^{l,i}\eta_\beta^{l,i}\rho^{l-2i} & \text{if } m \equiv n \pmod 2 \\ 0 & \text{o.w.} \end{cases}.$$

$\square$

Now the population risk becomes

$$g(A) = \frac{1}{2}\left\| \mathbb{E}_{\boldsymbol{x}\sim\mathcal{D}}\left[\boldsymbol{x}\boldsymbol{x}^\top\right] - \mathbb{E}_{\boldsymbol{z}\sim\mathcal{N}(0,I_{k\times k})}\left[\phi(A\boldsymbol{z})\phi(A\boldsymbol{z})^\top\right]\right\|^2$$
$$=\frac{1}{2}\sum_{i,j\in[d]}\left(\mathbb{E}_{\boldsymbol{z}\sim\mathcal{N}(0,I_{k_0\times k_0})}\phi((\boldsymbol{a}_i^*)^\top\boldsymbol{z})\phi((\boldsymbol{a}_j^*)^\top\boldsymbol{z}) - \mathbb{E}_{\boldsymbol{z}\sim\mathcal{N}(0,I_{k\times k})}\phi(\boldsymbol{a}_i^\top\boldsymbol{z})\phi(\boldsymbol{a}_j^\top\boldsymbol{z})\right)^2$$
$$\equiv\frac{1}{2}\sum_{i,j}\tilde{g}_{ij}(z_{ij}).$$

To simplify the notation, for a specific $i, j$ pair, we write $\hat{x} = \boldsymbol{a}_i^\top\boldsymbol{z}/\alpha$, $\alpha = \|\boldsymbol{a}_i\|$ and $\hat{y} = \boldsymbol{a}_j^\top\boldsymbol{z}/\beta$, where $\beta = \|\boldsymbol{a}_j\|$. Namely we have $(\hat{x}, \hat{y}) \sim \mathcal{N}(0, [[1, \rho]; [\rho, 1]])$, where $\rho = \cos\langle\boldsymbol{a}_i, \boldsymbol{a}_j\rangle$. Again, recall $\phi(\alpha\hat{x}) = \sum_{k \text{ odd}}\sigma_i h_i(\alpha\hat{x}) = \sum_{k \text{ odd}}\sigma_i h_i^\alpha(\hat{x})$.

$$\mathbb{E}_{\boldsymbol{z}\sim\mathcal{N}(0,I_{k\times k})}[\phi(\alpha\hat{x})\phi(\beta\hat{y})]$$
$$=\mathbb{E}\left[\sum_{m \text{ odd}}\sigma_m h_m^\alpha(\hat{x})\sum_{n \text{ odd}}\sigma_n h_n^\beta(\hat{y})\right]$$
$$=\sum_{m,n \text{ odd}}\sigma_m\sigma_n\mathbb{E}_S[h_m^\alpha(\hat{x})h_n^\beta(\hat{y})]$$
$$=\sum_{m \text{ odd}}\sigma_m\sum_{n\leq m \text{ odd}}\sigma_n\sum_{k=0}^{\lfloor \frac{n}{2} \rfloor}\eta_\alpha^{n,k}\eta_\beta^{n,k}\rho^{n-2k}$$

Therefore we could write out explicitly the coefficient for each term $\rho^k, k$ odd, as: $c_k = \sum_{n\geq k \text{ odd}}\sigma_n\eta_\alpha^{n,\frac{n-k}{2}}\eta_\beta^{n,\frac{n-k}{2}}(\sum_{m\geq n}\sigma_m)$. We have $\tilde{g}_{ij}(z_{ij}) = (\sum_{k \text{ odd}}c_k z_{ij}^k - \sum_{k \text{ odd}}c_k(z_{ij}^*)^k)^2$.

Now suppose $\sigma_i$ to have the same sign, and $\|\alpha_i\| \geq 1, \forall$ or $\|\alpha_i\| \leq 1, \forall i$, each coefficient $c_i \geq 0$. Therefore still the only stationary point for $g(Z)$ is $Z^*$.

# B OMITTED PROOFS FOR SAMPLE COMPLEXITY

## B.1 OMITTED PROOFS FOR RELATION ON APPROXIMATE STATIONARY POINTS

*Proof of Lemma 6.* We first review what we want to prove. For a matrix $A$ that satisfies $\epsilon$-approximate SOSP for Eqn. (5), we define $S_A = \nabla_Z\tilde{g}(AA^\top) - \sum_{i=1}^n\lambda_i X_i$. The conditions ensure that $A, \lambda, S_A$ satisfy:

$$\begin{cases} \text{Tr}(A^\top X_i A) = y_i, \\ \|S_A\tilde{\boldsymbol{a}}_i\|_2 \leq \epsilon\|\tilde{\boldsymbol{a}}_i\|_2, & \{\tilde{\boldsymbol{a}}_j\}_j \text{ span the column space of } A \\ \text{Tr}(B^\top D_A\nabla_A\mathcal{L}(A,\lambda)[B]) \geq -\epsilon\|B\|_F^2, & \forall B \text{ s.t. } \text{Tr}(B^\top X_i A) = 0. \end{cases} \qquad (9)$$

We just want to show $Z := AA^\top, \sigma := \lambda$, and $S := S_A$ satisfies the conditions for $\epsilon$-FOSP of Eqn. (6). Therefore, by going over the conditions, its easy to tell that all other conditions automatically apply and it remains to show $S_A \succeq -\epsilon I$.

By noting that $\nabla_A \mathcal{L}(A, \lambda) = 2S_A A$, one has:

$$\frac{1}{2}\operatorname{Tr}(B^\top D_A \nabla_A \mathcal{L}(A, \lambda)[B])$$

$$= \operatorname{Tr}(B^\top S_A B) + \operatorname{Tr}(B^\top D_A \nabla_Z \tilde{g}(AA^\top)[B]A) - \sum_{i=1}^{n} D_A \lambda_i[B] \operatorname{Tr}(B^\top X_i A)$$

(from Lemma 5 of Journée et al. (2008))

$$= \operatorname{Tr}(B^\top S_A B) + \operatorname{Tr}(AB^\top D_A \nabla_Z \tilde{g}(AA^\top)[B]) \tag{10}$$

(From Eqn. (9) we have $\operatorname{Tr}(B^\top X_i A) = 0$)

Notice that $A \in \mathbb{R}^{d \times k}$ and we have chosen $k = d$ for simplicity. We first argue when $A$ is rank-deficient, i.e. $\operatorname{rank}(A) < k$. There exists some vector $\boldsymbol{v} \in \mathbb{R}^k$ such that $A\boldsymbol{v} = 0$. Now for any vector $\boldsymbol{b} \in \mathbb{R}^d$, let $B = \boldsymbol{b}\boldsymbol{v}^\top$. Therefore $AB^\top = A\boldsymbol{v}\boldsymbol{b}^\top = 0$. From (10) we further have:

$$\frac{1}{2}\operatorname{Tr}(B^\top D_A \nabla_A \mathcal{L}(A, \lambda)[B])$$

$$= \operatorname{Tr}(B^\top S_A B) + \operatorname{Tr}(AB^\top D_A \nabla_Z \tilde{g}(AA^\top)[B])$$

$$= \operatorname{Tr}(\boldsymbol{v}\boldsymbol{b}^\top S_A \boldsymbol{b}\boldsymbol{v}^\top) = \|\boldsymbol{v}\|^2 \boldsymbol{b}^\top S_A \boldsymbol{b}$$

$$\geq -\epsilon/2\|B\|_F^2 \qquad \text{(from (9))}$$

$$= -\epsilon/2\|\boldsymbol{v}\|^2\|\boldsymbol{b}^\top\|^2$$

Therefore from the last three rows we have $\boldsymbol{b}^\top S_A \boldsymbol{b} \geq -\epsilon/2\|\boldsymbol{b}\|^2$ for any $\boldsymbol{b}$, i.e. $S_A \succeq -\epsilon/2 I_{d \times d}$. On the other hand, when $A$ is full rank, the column space of $A$ is the entire $\mathbb{R}^d$ vector space, and therefore $S_A \succeq -\epsilon I_{d \times d}$ directly follows from the second line of the $\epsilon$-SOSP definition.

$\square$

## B.2 DETAILED CALCULATIONS

Recall the population risk

$$g(A) \quad \equiv \quad \frac{1}{2}\left\|\mathbb{E}_{\boldsymbol{x} \sim \mathcal{D}}\left[\boldsymbol{x}\boldsymbol{x}^\top\right] - \mathbb{E}_{\boldsymbol{z} \sim \mathcal{N}(0, I_{k \times k})}\left[\phi(A\boldsymbol{z})\phi(A\boldsymbol{z})^\top\right]\right\|_F^2.$$

Write the empirical risk on observations as:

$$g_n(A) \quad \equiv \quad \frac{1}{2}\left\|\frac{1}{n}\sum_{i=1}^{n}\boldsymbol{x}_i\boldsymbol{x}_i^\top - \mathbb{E}_{\boldsymbol{z} \sim \mathcal{N}(0, I_{k \times k})}\left[\phi(A\boldsymbol{z})\phi(A\boldsymbol{z})^\top\right]\right\|_F^2.$$

**Claim 4.**
$$\nabla g(A) - \nabla g_n(A) = 2\mathbb{E}_{\boldsymbol{z}}\left[diag(\phi'(A\boldsymbol{z}))(X - X_n)\phi(A\boldsymbol{z})\boldsymbol{z}^\top\right],$$
where $X = \mathbb{E}_{\boldsymbol{x} \sim \mathcal{D}}[\boldsymbol{x}\boldsymbol{x}^\top]$, and $X_n = \frac{1}{n}\sum_{i=1}^{n}\boldsymbol{x}_i\boldsymbol{x}_i^\top$.

*Proof.*

$$\nabla g(A) - \nabla g_n(A) = \nabla(g(A) - g_n(A))$$

$$= \frac{1}{2}\nabla\left\langle X - X_n, X + X_n - 2\mathbb{E}_{\boldsymbol{z} \sim \mathcal{N}(0, I_{k \times k})}\left[\phi(A\boldsymbol{z})\phi(A\boldsymbol{z})^\top\right]\right\rangle$$

$$= \nabla\left\langle X_n - X, \mathbb{E}_{\boldsymbol{z} \sim \mathcal{N}(0, I_{k \times k})}\left[\phi(A\boldsymbol{z})\phi(A\boldsymbol{z})^\top\right]\right\rangle$$

Now write $S(A) = \phi(A\boldsymbol{z})\phi(A\boldsymbol{z})^\top$.

$$[S(A + \Delta A) - S(A)]_{ij}$$

$$= \phi(\boldsymbol{a}_i^\top \boldsymbol{z} + \Delta \boldsymbol{a}_i^\top \boldsymbol{z})\phi(\boldsymbol{a}_j^\top \boldsymbol{z} + \Delta \boldsymbol{a}_j^\top \boldsymbol{z}) - \phi(\boldsymbol{a}_i^\top \boldsymbol{z})\phi(\boldsymbol{a}_j^\top \boldsymbol{z})$$

$$= \phi'(\boldsymbol{a}_i^\top \boldsymbol{z})\Delta \boldsymbol{a}_i^\top \boldsymbol{z}\phi(\boldsymbol{a}_j^\top \boldsymbol{z}) + \phi'(\boldsymbol{a}_j^\top \boldsymbol{z})\Delta \boldsymbol{a}_j^\top \boldsymbol{z}\phi(\boldsymbol{a}_i^\top \boldsymbol{z}) + \mathcal{O}(\|\Delta A\|^2)$$

Therefore

$$
\begin{aligned}
&[S(A + \Delta A) - S(A)]_{i:} \\
&= \phi'(a_i^\top z)\Delta a_i^\top z\phi(Az)^\top + (\phi'(Az) \circ \Delta Az)^\top \phi(a_i^\top z) + \mathcal{O}(\|\Delta A\|^2)
\end{aligned}
$$

Therefore

$$
S(A + \Delta A) - S(A) = \mathrm{diag}(\phi'(Az))\Delta Az\phi(Az)^\top + \phi(Az)z^\top \Delta A^\top \mathrm{diag}(\phi'(Az)). \tag{11}
$$

And

$$
\begin{aligned}
&g(A + \Delta A) - g_n(A + \Delta A) - (g(A) - g_n(A)) \\
&= \langle X_n - X, \mathbb{E}_z\left[S(A + \Delta A) - S(A)\right]\rangle \\
&= \mathbb{E}_z\langle X_n - X, \mathrm{diag}(\phi'(Az))\Delta Az\phi(Az)^\top + \phi(Az)z^\top \Delta A^\top \mathrm{diag}(\phi'(Az))\rangle \\
&= 2\mathbb{E}_z\langle \mathrm{diag}(\phi'(Az))(X_n - X)\phi(Az)z^\top, \Delta A\rangle.
\end{aligned}
$$

Finally we have $\nabla g(A) - \nabla g_n(A) = 2\mathbb{E}_z\left[\mathrm{diag}(\phi'(Az))(X_n - X)\phi(Az)z^\top\right]$. □

**Claim 5.** *For arbitrary matrix B, the directional derivative of $\nabla g(A) - \nabla g_n(A)$ with direction B is:*

$$
\begin{aligned}
&D_A\nabla g(A)[B] - D_A\nabla g_n(A)[B] \\
&= 2\mathbb{E}_z\left[diag(\phi'(Az))(X_n - X)\phi'(Az) \circ (Bz)z^\top\right] \\
&\quad + 2\mathbb{E}_z\left[diag(\phi''(Az) \circ (Bz))(X_n - X)\phi(Az)z^\top\right]
\end{aligned}
$$

*Proof.*

$$
\begin{aligned}
&g(A + tB) \\
&= 2\mathbb{E}_z\left[\mathrm{diag}(\phi'(Az + tBz))(X_n - X)\phi(Az + tBz)z^\top\right] \\
&= 2\mathbb{E}_z\left[\mathrm{diag}(\phi'(Az) + t(Bz) \circ \phi''(Az))(X_n - X)(\phi(Az) + t\phi'(Az) \circ (Bz))z^\top\right] + \mathcal{O}(t^2)
\end{aligned}
$$

Therefore

$$
\begin{aligned}
&\lim_{t \to 0}\frac{g(A + tB) - g(A)}{t} \\
&= 2\mathbb{E}_z\left[\mathrm{diag}(\phi'(Az))(X_n - X)\phi'(Az) \circ (B^\top z)z^\top\right] \\
&\quad + 2\mathbb{E}_z\left[\mathrm{diag}(\phi''(Az) \circ (Bz))(X_n - X)\phi(Az)z^\top\right]
\end{aligned}
$$

□

### B.3 OMITTED PROOFS FOR OBSERVATION SAMPLE COMPLEXITY

*Proof of Lemma 3.* For each $x_i = \phi(Az_i), z_i \sim \mathcal{N}(0, I_{k \times k})$. Each coordinate $|x_{i,j}| = |\phi(a_j^\top z_i)| \le |a_j^\top z_i|$ since $\phi$ is 1-Lipschitz. [6]. Without loss of generality we assumed $\|a_j\| = 1, \forall j$, therefore $a_j^\top z \sim \mathcal{N}(0, I_{k \times k})$. For all $i \in [n], j \in [d]$ $|x_{i,j}| \le \log(nd/\delta)$ with probability $1 - \delta$.

Then by matrix concentration inequality ((Vershynin, 2010) Corollary 5.52), we have with probability $1 - \delta$: $(1 - \epsilon)X \preceq X_n \preceq (1 + \epsilon)X$ if $n \ge \Omega(d/\epsilon^2 \log^2(nd/\delta))$. Therefore set $n = \tilde{\Theta}(d/\epsilon^2 \log^2(1/\delta))$ will suffice. □

*Proof of Lemma 4.*

$$
\begin{aligned}
X_{ij} &= \mathbb{E}_{z \sim \mathcal{N}(0, I_{k \times k})}\phi(a_i^\top z)\phi(a_j^\top z) \\
&= \begin{cases} 0 & i \ne j \\ \mathbb{E}[\phi^2(a_i^\top z)] \le \frac{2}{\pi} & i = j \end{cases}
\end{aligned}
$$

---

[6] For simplicity, we analyze as if $\phi(0) = 0$ w.o.l.g. throughout this section, since the bias term is canceled out in the observation side with $\phi(A^* z)$ and the learning side with $\phi(Az)$.

Therefore $\|X\|_2 \leq \frac{2}{\pi}$. Together with Lemma 3, $\|X - X_n\| \leq \epsilon \frac{2}{\pi}$ w.p $1 - \delta$. Recall
$$\nabla g(A) - \nabla g_n(A) = 2\mathbb{E}_{\boldsymbol{z}} \left[ \operatorname{diag}(\phi'(A\boldsymbol{z}))(X - X_n)\phi(A\boldsymbol{z})\boldsymbol{z}^\top \right] := 2\mathbb{E}_{\boldsymbol{z}} G(\boldsymbol{z}),$$
where $G(\boldsymbol{z})$ is defined as $\operatorname{diag}(\phi'(A\boldsymbol{z}))(X - X_n)\phi(A\boldsymbol{z})\boldsymbol{z}^\top$. We have $\|G(z)\| \leq \|A\|\|\boldsymbol{z}\|^2\|X - X_n\|$.

$$
\begin{aligned}
\|\nabla g(A) - \nabla g_n(A)\|_2 &= 2\|\mathbb{E}_{\boldsymbol{z}}[G(\boldsymbol{z})]\| \\
&\leq 2\mathbb{E}_{\boldsymbol{z}}\|G(\boldsymbol{z})\| \\
&\leq 2\mathbb{E}_{\boldsymbol{z}}\|A\|\|\boldsymbol{z}\|^2\|X - X_n\| \\
&\leq 2\|A\|\epsilon\frac{2}{\pi}\mathbb{E}_{\boldsymbol{z}}\|\boldsymbol{z}\|^2 \\
&= 2\|A\|\epsilon d\frac{2}{\pi}
\end{aligned}
$$

For the directional derivative, we make the concentration bound in a similar way. Denote
$$D(\boldsymbol{z}) = \operatorname{diag}(\phi'(A\boldsymbol{z}))(X_n - X)\phi'(A\boldsymbol{z}) \circ (B\boldsymbol{z})\boldsymbol{z}^\top + \operatorname{diag}(\phi''(A\boldsymbol{z}) \circ (B\boldsymbol{z}))(X_n - X)\phi(A\boldsymbol{z})\boldsymbol{z}^\top.$$
$$\|D(\boldsymbol{z})\| \leq \|X_n - X\|_2\|B\|\|\boldsymbol{z}\|^2(1 + \|\boldsymbol{z}\|\|A\|).$$
Therefore $\|D_A\nabla g(A)[B] - D_A\nabla g_n(A)[B]\| \leq \mathcal{O}(\epsilon d^{3/2}\|A\|\|B\|)$ with probability $1 - \delta$. $\qquad\square$

## B.4 Omitted Proofs on Bounding Mini-Batch Size

Recall
$$\tilde{g}_{m,n}(A) \equiv \frac{1}{2} \left\| \frac{1}{n}\sum_{i=1}^n \boldsymbol{x}_i\boldsymbol{x}_i^\top - \frac{1}{m}\sum_{j=1}^m \phi(A\boldsymbol{z}_j)\phi(A\boldsymbol{z}_j)^\top \right\|_F^2.$$
Write $S_j(A) \equiv \phi(A\boldsymbol{z}_j)\phi(A\boldsymbol{z}_j)^\top$. Then we have
$$
\begin{aligned}
\tilde{g}_{m,n}(A) &= \frac{1}{2}\left\langle X_n - \frac{1}{n}\sum_{j=1}^m S_j(A), X_n - \frac{1}{m}\sum_{j=1}^m S_j(A) \right\rangle \\
&= \frac{1}{2m^2}\sum_{i,j}\langle S_i(A), S_j(A)\rangle - \frac{1}{n}\sum_{j=1}^m \langle S_j(A), X_n\rangle + \frac{1}{2}\|X_n\|_F^2
\end{aligned}
$$

On the other hand, our target function is:
$$
\begin{aligned}
g_n(A) &\equiv \frac{1}{2}\left\| \frac{1}{n}\sum_{i=1}^n \boldsymbol{x}_i\boldsymbol{x}_i^\top - \mathbb{E}_{\boldsymbol{z}\sim\mathcal{N}(0,I_{k\times k})}\left[\phi(A\boldsymbol{z})\phi(A\boldsymbol{z})^\top\right] \right\|_F^2 \\
&= \frac{1}{2}\|\mathbb{E}_S[S]\|_F^2 - \langle \mathbb{E}_S[S], X_n\rangle + \frac{1}{2}\|X_n\|_F^2
\end{aligned}
$$
Therefore $\mathbb{E}_S\tilde{g}_{m,n}(A) - g_n(A) = \frac{1}{2m}(\mathbb{E}_S\|S(A)\|_F^2 - \|\mathbb{E}_S S(A)\|_F^2)$.

**Claim 6.**
$$\nabla\mathbb{E}_S\tilde{g}_{m,n}(A) - \nabla g_n(A) = \frac{2}{m}\mathbb{E}_{\boldsymbol{z}}\left[\operatorname{diag}(\phi'(A\boldsymbol{z}))S(A)\phi(A\boldsymbol{z})\boldsymbol{z}^\top - \operatorname{diag}(\phi'(A\boldsymbol{z}))\mathbb{E}_S[S(A)]\phi(A\boldsymbol{z})\boldsymbol{z}^\top\right].$$

*Proof.*
$$
\begin{aligned}
&\langle \nabla\mathbb{E}_S\tilde{g}_{m,n} - \nabla g_n, \Delta A\rangle \\
&= \mathbb{E}_S\tilde{g}_{m,n}(A + \Delta A) + g_n(A + \Delta A) - (\mathbb{E}_S\tilde{g}_{m,n}(A) + g_n(A)) + \mathcal{O}(\|\Delta A\|^2) \\
&= \frac{1}{2m}\left(\mathbb{E}_S\|S(A + \Delta A)\|_F^2 - \mathbb{E}_S\|S(A)\|_F^2 - \|\mathbb{E}_S S(A + \Delta A)\|_F^2 + \|\mathbb{E}_S S(A)\|_F^2\right) + \mathcal{O}(\|\Delta A\|^2) \\
&= \frac{1}{m}\left(\mathbb{E}_S\langle S(A), S(A + \Delta A) - S(A)\rangle - \langle \mathbb{E}_S[S(A)], E_S[S(A + \Delta A) - S(A)]\rangle + \mathcal{O}(\|\Delta A\|^2)\right) \\
&= \frac{1}{m}\left(\langle \mathbb{E}_{\boldsymbol{z}}\langle S(A), \operatorname{diag}(\phi'(A\boldsymbol{z}))\Delta A\boldsymbol{z}\phi(A\boldsymbol{z})^\top\rangle - \langle\mathbb{E}_S[S(A)], \mathbb{E}_{\boldsymbol{z}}\operatorname{diag}(\phi'(A\boldsymbol{z}))\Delta A\boldsymbol{z}\phi(A\boldsymbol{z})^\top\rangle\right) \\
&\quad + \mathcal{O}(\|\Delta A\|^2) \qquad\qquad\qquad\qquad\qquad\qquad\qquad\qquad \text{(from Eqn. (11) and symmetry of } S) \\
&= \langle \frac{2}{m}\mathbb{E}_{\boldsymbol{z}}\left[\operatorname{diag}(\phi'(A\boldsymbol{z}))S(A)\phi(A\boldsymbol{z})\boldsymbol{z}^\top - \operatorname{diag}(\phi'(A\boldsymbol{z}))\mathbb{E}_S[S(A)]\phi(A\boldsymbol{z})\boldsymbol{z}^\top\right], \Delta A\rangle + \mathcal{O}(\|\Delta A\|^2)
\end{aligned}
$$

$\square$

Similarly to the derivation in the previous subsection, we again derive the bias in the directional derivative:

**Claim 7.** *For arbitrary matrix direction $B$,*

$$D_A \nabla \mathbb{E}_S \tilde{g}_{m,n}(A)[B] - D_A \nabla g_n(A)[B]$$
$$= \frac{2}{m} \mathbb{E}_{\boldsymbol{z}} \big[ diag(\phi''(A\boldsymbol{z}) \circ (B\boldsymbol{z}))(S(A) - \mathbb{E}_S S(A)) \phi(A\boldsymbol{z}) \boldsymbol{z}^\top$$
$$+ diag(\phi'(A\boldsymbol{z})) \left( (\phi'(A\boldsymbol{z}) \circ (B\boldsymbol{z})) \phi(A\boldsymbol{z})^\top - \mathbb{E}_{\boldsymbol{z}}[(\phi'(A\boldsymbol{z}) \circ (B\boldsymbol{z})) \phi(A\boldsymbol{z})^\top] \right) \phi(A\boldsymbol{z}) \boldsymbol{z}^\top$$
$$+ diag(\phi'(A\boldsymbol{z})) \left( \phi(A\boldsymbol{z})(\phi'(A\boldsymbol{z}) \circ (B\boldsymbol{z}))^\top - \mathbb{E}_{\boldsymbol{z}}[\phi(A\boldsymbol{z})(\phi'(A\boldsymbol{z}) \circ (B\boldsymbol{z}))^\top] \right) \phi(A\boldsymbol{z}) \boldsymbol{z}^\top$$
$$+ diag(\phi'(A\boldsymbol{z}))(S(A) - \mathbb{E}_S S(A))(\phi'(A\boldsymbol{z}) \circ (B\boldsymbol{z})) \boldsymbol{z}^\top \big]$$

### B.5 Omitted proof of the main theorem

*Proof of Lemma 7.* On one hand, suppose $Z$ is an $\epsilon$-FOSP property of $\tilde{g}$ in (6) along with the matrix $S$ and vector $\sigma$, we have:

$$\langle \nabla \tilde{g}(Z), Z - Z^* \rangle$$
$$= \langle S, Z - Z^* \rangle$$
$$\text{(since } Z - Z^* \text{ has 0 diagonal entries)}$$
$$\leq \|P_T(S)\|_2 \|P_{T \circ}(Z - Z^*)\|_F$$
$$\text{(}T \text{ is the tangent cone of PSD matrices at } Z\text{)}$$
$$\leq \|P_T(S)\|_2 \|Z - Z^*\|_F$$
$$= \max_j \{\tilde{\boldsymbol{a}}_j^\top S \tilde{\boldsymbol{a}}_j\} \|Z - Z^*\|_F$$
$$\text{(}\tilde{\boldsymbol{a}}_j \text{ is the basis of the column space of } Z\text{)}$$
$$\leq \epsilon \|Z - Z^*\| \tag{12}$$
$$\text{(from the definition of } \epsilon\text{-FOSP)}$$

On the other hand, from the definition of $\tilde{g}$, we have:

$$\langle Z - Z^*, \nabla \tilde{g}(Z) \rangle$$
$$= \sum_{ij} (z_{ij} - z_{ij}^*) \tilde{g}_{ij}'(z_{ij})$$
$$= \sum_{ij} (z_{ij} - z_{ij}^*)^2 \sum_{k \text{ odd}} \sigma_k^2 P_k(z_{ij}) \sum_{k \text{ odd}} \sigma_k^2 k z_{ij}^{k-1}$$
$$\geq \|Z - Z^*\|_F^2 \sigma_1^4 \tag{13}$$

Here polynomial $P_k(z_{ij}) \equiv (z_{ij}^k - (z_{ij}^*)^k)/(z - z^*)$ is always positive for $z \neq z^*$ and $k$ to be odd.

Therefore by comparing (12) and (13) we have $\epsilon \|Z - Z^*\|_F \geq \|Z - Z^*\|_F^2 \sigma_1^4$, i.e. $\|Z - Z^*\|_F \leq \mathcal{O}(\epsilon)$. $\square$

*Proof of Theorem 3.* From Theorem 31 from Ge et al. (2015), we know for small enough learning rate $\eta$, and arbitrary small $\epsilon$, there exists large enough $T$, such that Algorithm 1 generates an output $A^{(T)}$ that is sufficiently close to the second order stationary point for $f$. Or formally we have,

$$\begin{cases} \text{Tr}((A^{(T)})^\top X_i A^{(T)}) = y_i, \\ \|(\nabla_A f(A^{(T)}) - \sum_{i=1} \lambda_i X_i A^{(T)})_{:,j}\|_2 \leq \epsilon \min \|A_{j,:}\|_2, & \forall j \in [k] \\ \text{Tr}(B^\top D_A \nabla_A \mathcal{L}_f(A^{(T)}, \lambda)[B]) \geq -\epsilon \|B\|_2^2, & \forall B, s.t. \text{Tr}(B^\top X_i A) = 0 \end{cases}$$

$\mathcal{L}_f(A, \lambda) = f(A) - \sum_{i=1}^d \lambda_i(\text{Tr}(A^\top X_i A) - y_i)$. Let $\{\tilde{\boldsymbol{a}}_i = A^{(T)} \boldsymbol{r}_i\}_i^k$ to form the basis of the column vector space of $A^{(T)}$. Then the second line is a sufficient condition for the following: $\|\tilde{\boldsymbol{a}}_j^\top (\nabla_A f(A^{(T)}) - \sum_{i=1} \lambda_i X_i A^{(T)}) \boldsymbol{r}_j\|_2 \leq \epsilon, \forall j \in [k]$.

Now with the concentration bound from Lemma 5, suppose our batch size $m \geq \mathcal{O}(d^5/\epsilon)$, we have $\|\nabla_A g_n(A^{(T)}) - \nabla_A f(A^{(T)})\|_2 \leq \epsilon$, and $\|D_A \nabla_A g_n(A^{(T)})[B] - D_A \nabla_A f(A^{(T)})[B]\|_2 \leq \epsilon \|B\|_2$ for arbitrary $B$. Therefore again we get:

$$\begin{cases} \mathrm{Tr}((A^{(T)})^\top X_i A^{(T)}) = y_i \\ \|\tilde{\boldsymbol{a}}_j^\top (\nabla_A g_n(A^{(T)}) - \sum_{i=1} \lambda_i X_i A^{(T)}) \boldsymbol{r}_j\|_2 \leq 2\epsilon, & \forall j \in [k] \\ \mathrm{Tr}(B^\top D_A \nabla_A \mathcal{L}_{g_m}(A^{(T)}, \lambda)[B]) \geq -2\epsilon \|B\|_2^2, & \forall B, s.t.\, \mathrm{Tr}(B^\top X_i A) = 0 \end{cases}$$

Next we turn to the concentration bound from Lemma 4. Suppose we have when the sample size $n \geq \mathcal{O}(d^5/\epsilon^2 \log^2(1/\delta))$, $\|D_A \nabla_A g(A)[B] - D_A \nabla_A g_n(A)[B]\|_2 \leq \mathcal{O}(\epsilon \|B\|_2)$, and $\|\nabla g(A) - \nabla g_n(A)\|_2 \leq \mathcal{O}(\epsilon)$ with probability $1 - \delta$. Therefore similarly we get $A^{(T)}$ is an $\mathcal{O}(\epsilon)$-SOSP for $g(A) = \frac{1}{2} \left\| \sum_{i=0}^\infty \sigma_i^2 \left( (A^*(A^*)^\top)^{\circ i} - (AA^\top)^{\circ i} \right) \right\|_F^2$.

Now with Lemma 6 that connects the approximate stationary points, we have $Z := A^{(T)}(A^{(T)})^\top$ is also an $\epsilon$-FOSP of $\tilde{g}(Z) = \frac{1}{2} \left\| \sum_{i=0}^\infty \sigma_i^2 \left( (Z^*)^{\circ i} - Z^{\circ i} \right) \right\|_F^2$.

Finally with Lemma 7, we get $\|Z - Z^*\|_F \leq \mathcal{O}(\epsilon)$.

$\square$

