# OpenReview forum: "SGD Learns One-Layer Networks in WGANs"
_ICLR.cc/2020/Conference — Reject_

### Official Review · AnonReviewer3 · 2019-10-23
**Official Blind Review #3**

**Rating:** 3

**Review:**

In this paper, the authors attempt to prove that the Stochastic Gradient Descent-Ascent could converge to a global solution to the min-max problem of WGAN, in the setting of a one-layer generator and simple discriminator. They also show that the linear discriminator could be used to learn the marginal distributions of each coordinate, while a quadratic one could obtain joint distributions of every two coordinates. Since the linear discriminator and the quadratic one could be solved in one step Gradient Ascent, the author applied the standard analysis method to reveal the property of the Gradient Descent method. Experiments are also carried out to justify their theory that the WGAN could recover the distribution.

However, the most significant drawback of this paper is that the settings for the discriminator are too simple, which leads to the following two problems: 1) Revealing the joint distributions of two coordinates is still much weaker than the desired result of recovering the true distribution of the data. 2) The analysis of this paper could not be extended to a complex discriminator since it would be suffered from the training error propagation in the Gradient Ascent step, instead of getting an accurate solution for the Gradient Ascent step.

Therefore, more explanations are desired to be given to bound the error propagation and what will the complimentary discriminator learn from the data distribution.

**Experience Assessment:**

I have read many papers in this area.

**Review Assessment: Checking Correctness Of Derivations And Theory:**

I assessed the sensibility of the derivations and theory.

**Review Assessment: Checking Correctness Of Experiments:**

I assessed the sensibility of the experiments.

**Review Assessment: Thoroughness In Paper Reading:**

I read the paper thoroughly.

---

> ### Author Response · Authors · 2019-11-13
> **Response to Review #3**
>
> Thank you for your reviews. Unfortunately there is a significant misunderstanding of our contributions. We will try to clarify some concerns here and hope it will justify our contributions more clearly.
>
> We want to emphasize our contributions first.
> 1. To begin with, the global convergence of gradient descent-ascent in the GAN setting has not been extensively studied. We provide the $\textbf{first result}$ to show $\textbf{convergence to global equilibrium points}$ for $\textbf{non-linear generators}$. The difficulty in analyzing gradient descent-ascent is twofold: the generator dynamics and discriminator dynamics. On the discriminator side, our choice of quadratic discriminator not only simplifies the dynamics but also has sufficient discriminating power (we will justify it  below). On the generator side, its minimization problem is non-convex, and therefore our convergence result to global equilibria is highly non-trivial. Our primary contribution in gradient descent-ascent analysis is to choose a proper discriminator set and to understand the generator dynamics.
>
> 2. For the generator class we are considering, we proved the quadratic discriminator both $\textbf{simplifies the gradient ascent dynamics}$ and $\textbf{attains a nearly optimal sample complexity}$ (see point 3 below). Had we chosen to use a more complex discriminator, even if the maximization step were tractable, this would increase the sample complexity, potentially to a non-parametric rate (Feizi et al., 2017; Bai et al., 2018).
>
> 3. Our sample analysis also matches the upper bound of $O(1/\epsilon^2)$ on dependence of the error $\epsilon$ provided in (Wu et al., 2019). This is also a side proof that with WGAN we could $\textbf{learn one-layer generator}$ via $\textbf{appropriate discriminator class at a parametric rate}$.
>
> Next we justify our choice of discriminator class.
> We want to emphasize that our goal is to show that SGD learns the ground truth generating distribution, with minimal requirements for the discriminator class.
>
> Our choice of discriminator class, quadratic discriminators, already $\textbf{has sufficient distinguishing power}$ to learn the family of distributions parametrized by our generator class. As shown in Theorem 3, the quadratic discriminator class is sufficient to learn the optimal generator. In fact using a larger discriminator family will only make the learning harder by increasing the sample complexity; see (Feizi et al., 2017; Bai et al., 2018) for a discussion of the importance of appropriately constraining the discriminator class to attain parametric sample complexity. Our choice of small discriminator class is a strength, not a weakness.
>
> When more complex discriminators are necessary (on studying more complicated generators for future work), we believe the discriminator dynamics can be analyzed using recent developments in the training of neural networks for classification problems (e.g. NTK results). However, this is not the focus of our paper since we are learning to recover one-layer generator, which does not need a complex discriminator.
>
> Finally we clarify some other points you’ve raised.
> Q: “revealing the joint distributions of two coordinates is still much weaker than the desired result of recovering the true distribution of the data”
> A: We prove in Theorem 1 that the gradient descent-ascent recovers the optimal generator, so we do achieve the desired result of recovering the true distribution of the data.
>
> Q: “more complex discriminator will cause train error propagation”
> A: As we have shown in Theorem 1, it is unnecessary to have a complex discriminator for our generator architecture.
>
> For more complex discriminator architectures, we believe it is possible to apply NTK results on the discriminator to analyze the discriminator dynamics. However, this is not the focus of our paper since we are learning to recover one-layer generator, which does not need a complex discriminator.
>
> Reference:
> (Feizi et al. 2017) Feizi, S., Farnia, F., Ginart, T., & Tse, D. (2017). Understanding GANs: the LQG setting. arXiv preprint arXiv:1710.10793.
> (Bai et al. 2018) Bai, Y., Ma, T., & Risteski, A. (2018). Approximability of discriminators implies diversity in GANs. arXiv preprint arXiv:1806.10586.
> (Wu et al. 2019) Wu, S., Dimakis, A. G., & Sanghavi, S, “Learning Distributions Generated by One-Layer ReLU Networks”, NeurIPS 2019

---

### Official Review · AnonReviewer1 · 2019-10-23
**Official Blind Review #1**

**Rating:** 6

**Review:**

I have read the authors response. In the response the authors clarified the contributions of this paper. I agree with the authors that the analysis of gradient descent-ascent is a difficult problem, and the optimization results given in this paper is a contribution of importance. Because of this I have improved my score.

However, I do not agree with the authors that studying quadratic discriminators instead of more complicated ones should be considered as a contribution instead of drawback. In my opinion, as long as the focus is on WGAN, results involving standard neural networks are still more desired compared with the results in this submission. For example, similar results for a neural network discriminator might be even more impactful, because the optimization problem is even more difficult. Therefore I still consider the simple discriminator and generator as a weak point of this paper.


======================================================================================================

This paper studies the training of WGANs with stochastic gradient descent. The authors show that for one-layer generator network and quadratic discriminator, if the target distribution is modeled by a teacher network same as the generator, then stochastic gradient descent-ascent can learn this target distribution in polynomial time. The authors also provide sample complexity results.

The paper is well-written and the theoretical analysis seems to be valid and complete. However, I think the WGANs studied in this paper are simplified too much that the analysis can no longer capture the true nature of WGAN training.

First, the paper only studies linear and quadratic discriminators. This is not very consistent with the original intuition of WGAN, which is to use the worst Lipschitz continuous neural network to approximate the worst function in the set of all Lipschitz continuous functions in the definition of Wasserstein distance. When the discriminator is as simple as linear or quadratic functions, there is pretty much no “Wasserstein” in the optimization problem.

Moreover, the claim that SGD learns one-layer networks can be very misleading. In fact what is a “one-layer” neural network?
- if the authors meant “two-layer network” or “single hidden layer network”, then this is not true. Because as far as I can tell, the model $x = B \phi(A z)$ is much more difficult than the model $x = \phi(A z)$. The former is a standard single hidden layer network which is non-convex, while the latter is essentially a linear model especially when \phi is known.
- if the authors meant “a linear model with elementwise monotonic transform”, then I would like to suggest that a more appropriate name should be used to avoid unnecessary confusion.

As previously mentioned, the discriminators are too simple to approximate the Wasserstein distance, and therefore in general it should not be possible to guarantee recovery of the true data distribution. However, in this paper it is still shown that certain true distributions can be learned. This is due to the extremely simplified true model. In fact, even if the activation function $\phi$ is unknown, it seems that one can still learn $A^* (A^*)^\top$ well (for example, by Kendall’s tau).


**Experience Assessment:**

I have published one or two papers in this area.

**Review Assessment: Checking Correctness Of Derivations And Theory:**

I assessed the sensibility of the derivations and theory.

**Review Assessment: Checking Correctness Of Experiments:**

I assessed the sensibility of the experiments.

**Review Assessment: Thoroughness In Paper Reading:**

I read the paper at least twice and used my best judgement in assessing the paper.

---

> ### Author Response · Authors · 2019-11-13
> **Response to Review #1**
>
> Thank you for your reviews. Unfortunately there is a significant misunderstanding of our contributions. We will try to clarify some concerns here and hope it will justify our contributions more clearly.
>
> We want to emphasize our contributions first.
> 1. To begin with, the global convergence of gradient descent-ascent in the GAN setting has not been extensively studied. We provide the $\textbf{first result}$ to show $\textbf{convergence to global equilibrium points}$ for $\textbf{non-linear generators}$. The difficulty in analyzing gradient descent-ascent is twofold: the generator dynamics and discriminator dynamics. On the discriminator side, our choice of quadratic discriminator not only simplifies the dynamics but also has sufficient discriminating power (we will justify it  below). On the generator side, its minimization problem is non-convex, and therefore our convergence result to global equilibria is highly non-trivial. Our primary contribution in gradient descent-ascent analysis is to choose a proper discriminator set and to understand the generator dynamics.
>
> 2. For the generator class we are considering, we proved the quadratic discriminator both $\textbf{simplifies the gradient ascent dynamics}$ and $\textbf{attains a nearly optimal sample complexity}$ (see point 3 below). Had we chosen to use a more complex discriminator, even if the maximization step were tractable, this would increase the sample complexity, potentially to a non-parametric rate (Feizi et al., 2017; Bai et al., 2018).
>
> 3. Our sample analysis also matches the upper bound of $O(1/\epsilon^2)$ on dependence of the error $\epsilon$ provided in (Wu et al., 2019). This is also a side proof that with WGAN we could $\textbf{learn one-layer generator}$ via $\textbf{appropriate discriminator class at a parametric rate}$.
>
> Next we justify our choice of discriminator class.
> We want to emphasize that our goal is to show that SGD learns the ground truth generating distribution, with minimal requirements for the discriminator class.
>
> Our choice of discriminator class, quadratic discriminators, already $\textbf{has sufficient distinguishing power}$ to learn the family of distributions parametrized by our generator class. As shown in Theorem 3, the quadratic discriminator class is sufficient to learn the optimal generator. In fact using a larger discriminator family will only make the learning harder by increasing the sample complexity; see (Feizi et al., 2017; Bai et al., 2018) for a discussion of the importance of appropriately constraining the discriminator class to attain parametric sample complexity. Our choice of small discriminator class is a strength, not a weakness.
>
> When more complex discriminators are necessary (on studying more complicated generators for future work), we believe the discriminator dynamics can be analyzed using recent developments in the training of neural networks for classification problems (e.g. NTK results). However, this is not the focus of our paper since we are learning to recover one-layer generator, which does not need a complex discriminator.
>
> Finally we clarify some other points you’ve raised.
> Q: “what is one-layer generator” & “it can be learned easily”
> A: By one-layer generator we mean the second case as you have suggested. This terminology is also used in some prior work, for instance in (Wu et al., 2019). As we have emphasized, our goal is not just to learn the one-layer generator by any method, but to understand the dynamics of gradient descent-ascent with WGAN on learning the distribution. We also demonstrate the near optimal sample complexity when learning with WGAN. Even though the generator is a simple formulation, this work still provides the first result on successful learning a non-linear generator with WGAN setting.
>
> Reference:
> (Feizi et al. 2017) Feizi, S., Farnia, F., Ginart, T., & Tse, D. (2017). Understanding GANs: the LQG setting. arXiv preprint arXiv:1710.10793.
> (Bai et al. 2018) Bai, Y., Ma, T., & Risteski, A. (2018). Approximability of discriminators implies diversity in GANs. arXiv preprint arXiv:1806.10586.
> (Wu et al. 2019) Wu, S., Dimakis, A. G., & Sanghavi, S, “Learning Distributions Generated by One-Layer ReLU Networks”, NeurIPS 2019

---

### Official Review · AnonReviewer2 · 2019-10-23
**Official Blind Review #2**

**Rating:** 3

**Review:**

The authors provide a long text to justify their contributions and I have read it thoroughly. Unfortunately, I find the responses don't really address my concerns.

My major concern is that I cannot understand how quadratic discriminator can be treated as WGAN. The authors replied that the regularization considered in the paper might be treated as Lipschitz constraint for bounded data sets. However, the data sets can’t be bounded because in the paper, the authors consider a special case where the data sets generated from a teacher network where the input is Gaussian noise. Moreover, the authors said that they would add an explanation of this important point in the revision but I haven’t found any revision yet.

My another concern is that why the authors don’t study the two layer network discriminator. The authors replied that the choice of discriminator is designed in tandem with the choice of generator.  If they use a standard two layer ReLU network as discriminator, this would hurt the sample complexity. I partly agree with that it will be nice if we can design a better discriminator according to the different choice of generator. However, it will be more convincing to show the convergence of WGAN if the authors consider NN discriminator rather than quadratic discriminator which hardly be used in GAN.

==================================================================================================
I found this paper over claims its contribution a lot, which is quite misleading. The title of this work is SGD LEARNS ONE-LAYER NETWORKS IN WGANS. And the authors claim that they analyze the convergence of stochastic gradient descent ascent for Wasserstein GAN on learning a single layer generator network. But actually this paper only considers two kinds of simplified discriminators: a (rectified) linear discriminator and quadratic discriminator, which are very different from WGAN used in practice. The analysis of two special cases are hard to be extended to the analysis of WGAN and thus can hardly help to explain why WGAN is successfully trained by SGD in practice.

In section 3, the authors consider the rectified linear discriminator, which is quite similar to the standard two layer network with relu activation but the first layer is fixed. The authors prove that the generator can learn the marginal distribution but may not learn the joint distribution. In the beginning of section 4, the authors explain that this is because there is no interaction between different coordinates of the random vector. To learn joint distribution, the authors extend the linear discriminator to the quadratic discriminator and think of it as a natural idea.

For the rectified linear discriminator, the regularization of the discriminator is the norm the output layer of discriminator which can be related to the Lipschitz constraint in WGAN. But for quadratic discriminator, I cannot understand how this setting can be treated as WGAN without further explanation from the authors.

I wonder why this work doesn’t consider the standard two layer network discriminator which also has the interaction between different coordinates in the first layer.


**Experience Assessment:**

I have read many papers in this area.

**Review Assessment: Checking Correctness Of Derivations And Theory:**

I assessed the sensibility of the derivations and theory.

**Review Assessment: Checking Correctness Of Experiments:**

I assessed the sensibility of the experiments.

**Review Assessment: Thoroughness In Paper Reading:**

I read the paper thoroughly.

---

> ### Author Response · Authors · 2019-11-13
> **Response to Review #2**
>
> Thank you for your reviews. Unfortunately there is a significant misunderstanding of our contributions. We will try to clarify some concerns here and hope it will justify our contributions more clearly.
>
> We want to emphasize our contributions first.
> 1. To begin with, the global convergence of gradient descent-ascent in the GAN setting has not been extensively studied. We provide the $\textbf{first result}$ to show $\textbf{convergence to global equilibrium points}$ for $\textbf{non-linear generators}$. The difficulty in analyzing gradient descent-ascent is twofold: the generator dynamics and discriminator dynamics. On the discriminator side, our choice of quadratic discriminator not only simplifies the dynamics but also has sufficient discriminating power (we will justify it  below). On the generator side, its minimization problem is non-convex, and therefore our convergence result to global equilibria is highly non-trivial. Our primary contribution in gradient descent-ascent analysis is to choose a proper discriminator set and to understand the generator dynamics.
>
> 2. For the generator class we are considering, we proved the quadratic discriminator both $\textbf{simplifies the gradient ascent dynamics}$ and $\textbf{attains a nearly optimal sample complexity}$ (see point 3 below). Had we chosen to use a more complex discriminator, even if the maximization step were tractable, this would increase the sample complexity, potentially to a non-parametric rate (Feizi et al., 2017; Bai et al., 2018).
>
> 3. Our sample analysis also matches the upper bound of $O(1/\epsilon^2)$ on dependence of the error $\epsilon$ provided in (Wu et al., 2019). This is also a side proof that with WGAN we could $\textbf{learn one-layer generator}$ via $\textbf{appropriate discriminator class at a parametric rate}$.
>
> Next we justify our choice of discriminator class.
> We want to emphasize that our goal is to show that SGD learns the ground truth generating distribution, with minimal requirements for the discriminator class.
>
> Our choice of discriminator class, quadratic discriminators, already $\textbf{has sufficient distinguishing power}$ to learn the family of distributions parametrized by our generator class. As shown in Theorem 3, the quadratic discriminator class is sufficient to learn the optimal generator. In fact using a larger discriminator family will only make the learning harder by increasing the sample complexity; see (Feizi et al., 2017; Bai et al., 2018) for a discussion of the importance of appropriately constraining the discriminator class to attain parametric sample complexity. Our choice of small discriminator class is a strength, not a weakness.
>
> When more complex discriminators are necessary (on studying more complicated generators for future work), we believe the discriminator dynamics can be analyzed using recent developments in the training of neural networks for classification problems (e.g. NTK results). However, this is not the focus of our paper since we are learning to recover one-layer generator, which does not need a complex discriminator.
>
> Finally we clarify some other points you’ve raised.
> Q: “For quadratic discriminator, I cannot understand how this setting can be treated as WGAN”
> A: For quadratic discriminator, the square norm regularizer enforces that the Lipschitz constant of the discriminator is upper bounded by 1, for bounded data sets. We will add an explanation of this important point in the revision.
>
> Q: “why not study the two layer network discriminator”
> A: As we explained above, the choice of discriminator is designed in tandem with the choice of generator. If we use a standard two layer ReLU network as discriminator, this would hurt the sample complexity.
>
> Reference:
> (Feizi et al. 2017) Feizi, S., Farnia, F., Ginart, T., & Tse, D. (2017). Understanding GANs: the LQG setting. arXiv preprint arXiv:1710.10793.
> (Bai et al. 2018) Bai, Y., Ma, T., & Risteski, A. (2018). Approximability of discriminators implies diversity in GANs. arXiv preprint arXiv:1806.10586.
> (Wu et al. 2019) Wu, S., Dimakis, A. G., & Sanghavi, S, “Learning Distributions Generated by One-Layer ReLU Networks”, NeurIPS 2019

---

### Decision · Program_Chairs · 2019-12-19

**Decision:**

Reject

**Comment:**

This article studies convergence of WGAN training using SGD and generators of the form $\phi(Ax)$, with results on convergence with polynomial time and sample complexity under the assumption that the target distribution can be expressed by this type of generator. This expands previous work that considered linear generators. An important point of discussion was the choice of the discriminator as a linear or quadratic function. The authors' responses clarified some of the initial criticism, and the scores improved slightly. Following the discussion, the reviewers agreed that the problem being studied is a difficult one and that the paper makes some important contributions. However, they still found that the considered settings are very restrictive, maintaining that quadratic discriminators would work only for the very simple type of generators and targets under consideration. Although the article makes important advances towards understanding convergence of WGAN training with nonlinear models, the relevance of the contribution could be greatly enhanced by addressing / discussing the plausibility or implications of the analysis in a practical setting, in the best case scenario addressing a more practical type of neural networks.